# Hepatitis B virus X protein (HBx)-mediated immune modulation and prognostic model development in hepatocellular carcinoma

Jianhua Zhong[1☯], Yuetong Li[1☯], Yang Liu[2], Jie Qiao[2], Yiling Wu[2], Xinyi Kong[2], Miao Qi[2], Yiqi Lin[1], Yaqi Yao[1], Ying Jin[1], Changlong Bi[1]*, Aixia Zhai[2]*

**1** Department of Endocrinology, The Eighth Affiliated Hospital, Sun Yat-sen University, Shenzhen, China, **2** Department of Laboratory Medicine, The Eighth Affiliated Hospital, Sun Yat-sen University, Shenzhen, China

☯ These authors contributed equally to this work.
* aixiazhai@126.com, zhaiaix@mail.sysu.edu.cn (AZ); BCL163@163.com, bichlong@mail.sysu.edu.cn (CB)

## Abstract

Hepatitis B virus (HBV) X protein (HBx) is critical in hepatocellular carcinoma (HCC) development, but its influence on tumor immunity and the tumor microenvironment (TME) remains unclear. This study aimed to construct a prognostic model based on HBx-related genes and explore their relationship with immune infiltration and immunotherapy response. Through transcriptome sequencing of our HBx-expressing HepG2 cells and analysis of HCC patient data from the cancer genome atlas (TCGA) and genotype-tissue expression (GTEx), we identified seven HBx-related genes, nuclear VCP-like (NVL), WD repeat domain 75 (WDR75), NOP58 nucleolar protein (NOP58), Brix domain-containing protein 1 (BRIX1), deoxynucleotidyltransferase terminal interacting protein 2 (DNTTIP2), MKI67 FHA domain interacting nucleolar phosphoprotein (NIFK), and ribosome production factor 2 (RPF2), associated with poor prognosis. LASSO Cox regression narrowed these to four key genes (BRIX1, RPF2, DNTTIP2, and WDR75), which were used to develop a prognostic riskscore signature. High-risk patients exhibited lower survival rates, decreased infiltration of anti-tumor immune cells, poorer responses to immunotherapy, and increased immune evasion. Among the four genes, DNTTIP2 showed higher expression in single-cell data, was linked to migration inhibitory factor (MIF) signaling, and may play a pivotal role in shaping an immunosuppressive TME. Elevated DNTTIP2 expression was confirmed in HBx-expressing HepG2 cells and HBV-infected HCC samples. This study highlights a novel HBx-related four-gene prognostic model that predicts clinical outcomes, immune infiltration, and immunotherapy response, offering insights into HCC progression and potential therapeutic targets.

**Data availability statement:** Gene expression data were obtained from transcriptome sequencing of HepG2 cell samples transfected with the HBx plasmid and control plasmid. The dataset has been deposited in the GEO database under accession number GSE276530. The TCGA gene expression data and related clinical characteristics of HCC patients were downloaded from the UCSC Xena platform (https://xenabrowser.net/datapages), including 374 HCC samples and 50 normal samples. Single-cell transcriptome data were obtained from the GEO database under accession number GSE202642, including 7 HBV-related HCC samples and 4 adjacent liver tissue samples. All custom scripts and code used for data processing and analysis in this study are publicly available from Zenodo at https://doi.org/10.5281/zenodo.15638899. Other relevant data are within the paper and its Supporting Information files.

**Funding:** This work was supported by the National Natural Science Foundation of China (No.82072267), Shenzhen Science and Technology Innovation Program (No. JCYJ20240813150644056), Futian Healthcare Research Project (No.FTWS013), Sun Yat-Sen Eighth Affiliated Hospital Clinical Research Program (No.ZDBY-IIT-202304-055). Aixia Zhai received each award. The sponsors or funders didn't play any role in this study.

**Competing interests:** The authors have declared that no competing interests exist.

## Introduction

Hepatocellular carcinoma (HCC) is one of the most prevalent malignant tumors and the third leading cause of cancer-related mortality worldwide [1]. Approximately 50% to 80% of HCC cases are associated with chronic hepatitis B virus (HBV) infection [2,3]. HBV infection represents a significant global health threat, affecting 300 million individuals with chronic infections and resulting in up to one million deaths annually [4]. The virus can promote hepatocarcinogenesis through various mechanisms. A critical molecular pathway in liver cancer development is HBV DNA integration, which leads to chromosomal instability and mutations, resulting in the abnormal expression of oncogenes and tumor suppressor genes [5–8]. The HBV X protein (HBx) disrupts cellular signaling pathways and transcription, thereby impacting the cell cycle and cellular growth [9]. Additionally, the immune response and chronic inflammation triggered by HBV infection contribute to hepatocyte damage and liver fibrosis, further advancing the development of liver cancer [10–12].

HBx, encoded by the HBV X gene, is a principal viral protein that promotes hepatocarcinogenesis by enhancing the transcription of covalently closed circular DNA and driving viral replication, playing a critical role in HCC development and progression [13–15]. Research has demonstrated that the enhanced expression of various proto-oncogenes, such as c-myc, c-fos, and c-jun, is linked to the transcriptional transactivation function of HBx. Rawal et al. have found that HBx-positive liver cancer cells exhibited increased migration and invasion capabilities via endothelial-derived transforming growth factor-β (TGF-β) and facilitated epithelial-mesenchymal transition (EMT) through cluster of differentiation (CD) 133 expression, thereby enhancing tumor aggressiveness [16]. Moreover, HBx can promote hepatocarcinogenesis through epigenetic regulation mechanisms, including DNA methylation, histone modifications, and non-coding RNA expression [17]. Therefore, gaining a deeper understanding of the regulatory mechanisms of HBx in HCC is essential.

Immune abnormalities resulting from HBV infection are primary contributors to chronic liver damage. Prolonged chronic inflammation and alterations in the immune microenvironment ultimately facilitate the development of HCC. For instance, HBV-specific CD8+ T cells are essential for the clearance of HBV. However, in patients with chronic HBV infection, the functionality of these cells is compromised, characterized by limited proliferation and increased expression of inhibitory receptors such as programmed cell death protein 1 (PD1) and cytotoxic T lymphocyte-associated antigen 4 (CTLA4) [18–20]. Research has demonstrated that regulatory T (Tregs) cells are enriched in patients with HBV-related HCC, exhibiting an upregulation of inhibitory receptors that suppress anti-tumor immunity, leading to a higher incidence of immune exhaustion than other HCC types [21,22]. In circulating CD4+ T cells and γδ T cells from HBV-related HCC patients, the expression of inhibitory receptors such as T cell immune receptor with Ig and ITIM domains (TIGIT) and T cell immunoglobulin and mucin domain-containing protein 3 (TIM3) is elevated [23]. M2 macrophages, generally regarded as pro-tumorigenic, play a significant role in sustaining cancer stem cells through the M2 polarization of tumor-associated

macrophages in HBV-related HCC [24]. Tumor-infiltrating immune cells influence tumor progression, and understanding the characteristics of the tumor microenvironment (TME) will aid in developing targeted immunotherapy strategies [25]. The specific role of HBx in the TME of HBV-related HCC remains unclear, making further elucidation essential for enhancing patient prognosis. In this regard, recent studies have highlighted the potential impact of genetic polymorphisms, such as the ALDH2 Glu504Lys mutation, in shaping the susceptibility to HBV-related diseases and influencing the immune response in East Asian populations [26]. Furthermore, natural product-based antiviral candidates, such as baicalein and its glucuronide derivative baicalin, have gained attention for their antiviral properties and may offer promising therapeutic strategies for HBV-related diseases and HCC [27].

Despite advancements in the treatment of HCC through immunotherapy and targeted therapy, challenges persist, including limited efficacy and tumor resistance. This highlights the urgent need for effective biomarkers to monitor HCC and inform treatment decisions [28]. This study integrated unpublished transcriptomic data from our research group with publicly available databases to identify the key HBx-related genes through bulk and single-cell transcriptomic analyses. A novel riskscore signature was constructed using LASSO regression, demonstrating effectiveness in predicting prognosis, immunotherapy response, and drug selection. Additionally, cell-cell communication analysis suggested that tumor cells may regulate M2 macrophages and Tregs through the macrophage migration inhibitory factor (MIF) signaling, contributing to the formation of an immunosuppressive TME, with deoxynucleotidyltransferase terminal interacting protein 2 (DNTTIP2) potentially playing a crucial role. The findings of this study provide new insights into the immunoregulatory mechanisms of HBx in HBV-related HCC and suggest new directions for treatment strategies.

## Materials and methods

### Cell culture and transfection

The HepG2 cells, the culture and transfection methods were the same as previously described [29]. HepG2 cells were cultured in DMEM medium containing 10% fetal bovine serum and 1% penicillin-streptomycin, and incubated at 37°C in a humidified atmosphere with 5% $CO_2$. Cells were digested with trypsin and seeded into well plates. On the following day, when the cells reached 60–70% confluence, the medium was replaced with serum-free medium. According to the instructions for Lipofectamine 2000 (Invitrogen, USA), the transfection plasmids were diluted separately in serum-free medium. The cells were incubated at 37°C for 5 hours, after which the medium was replaced with complete medium and incubation continued until further experiments. The HBx plasmid was constructed by HBV X (NC_003977.2) gene into the pmCherry-N1 vector obtained from the MiaoLing Plasmid Platform (Wuhan, China). The plasmid constructs were sequence-verified to confirm the absence of mutations or frameshifts.

### Gene expression data collection

Gene expression data was from transcriptome sequencing of HepG2 cell samples transfected with the HBx plasmid and control plasmid. After uploading to the GEO database, the dataset was assigned the accession number GSE276530. The dataset GSE276530 consists of RNA sequencing data from six HepG2 cell samples, with three transfected with control plasmids and three transfected with HBx plasmids. The purpose of this experiment was to identify differentially expressed genes in HepG2 cells overexpressing HBx, in order to further analyze genes associated with poor prognosis in HBV-related HCC. HepG2 cells serve as a representative model for liver cancer research, and HBx is a key protein produced by HBV. The differential gene expression data obtained from this study will aid in understanding the molecular mechanisms underlying the poor prognosis of HBV-infected HCC patients. The cancer genome atlas (TCGA) gene expression data and related clinical characteristics of HCC patients were downloaded from UCSC Xena (https://xenabrowser.net/datapages), including 374 HCC samples and 50 normal samples. The single-cell transcriptome data was obtained from the GEO database [30] (GSE202642), including 7 HBV-related HCC samples and 4 adjacent liver tissue samples.

### Identification of differentially expressed genes (DEGs)

The DESeq2, edgeR, and limma R packages were used to obtain DEGs from the gene expression profiles. All identified DEGs met the criteria of P<0.05 and |log$_2$ (Fold-change)|>0.5. Genes consistently identified as DEGs by all three R packages were selected for further analysis. The ggplot2 and tinyarray R packages were used to visualize the identified DEGs.

### Functional enrichment analysis

The clusterProfiler [31] R package was used to perform Gene Ontology (GO) and KEGG (Kyoto Encyclopedia of Genes and Genomes) pathway enrichment analyses on the DEGs, with pathways showing P<0.05 considered statistically significant. The ggplot2 and tinyarray R packages were used to visualize the result.

### Analysis of protein-protein interactions (PPI)

The overlapping DEGs were analyzed for interaction information based on the STRING database [32] (https://string-db.org/), with the minimum required interaction score set to medium confidence (0.400). The interactions were visualized to obtain the PPI network using Cytoscape [33].

### Identification of hub genes

The PPI network was visualized with Cytoscape 3.9.0, and the significant sub-network module, was identified using the MCODE plugin (MCC algorithm). Survival analysis was performed on HCC samples from the TCGA data and the GEPIA database (http://gepia.cancer-pku.cn/detail.php), and key molecules with significant prognostic value were identified as hub genes.

### Immune cell infiltration analysis

Through single-sample gene-set enrichment analysis (ssGSEA), we estimated the abundance of 24 different types of immune cell infiltration based on gene expression profiles. The ESTIMATE algorithm was employed to calculate tumor purity, immune and stromal scores. Group comparisons were performed via the Wilcoxon rank-sum test. Spearman correlation analysis was conducted to explore the relationships between hub genes, risk score, and immune cells. Additionally, to simulate tumor immune evasion mechanisms and predict potential responses to tumor immunotherapy, we applied the tumor immune dysfunction and exclusion (TIDE) [34] algorithm and the immunopheno score (IPS) [35] algorithm.

### Generation of prognostic signatures

Based on the combined role of these 7 genes in HCC progression, we constructed a riskscore signature to comprehensively evaluate their impact on patient prognosis, TME immune cell infiltration, and immune response. The least absolute shrinkage and selection operator (LASSO) Cox regression was employed to reduce dimensionality and select the most robust markers for building the riskscore signature. The optimal penalty parameter λ was determined through 10-fold cross-validation.

The riskscore signature is defined as follows:

$$riskscore = \sum_{i=1}^{n} Coefi * Expri$$

Coefi represents the coefficients obtained from the LASSO Cox regression, and Expri represents the expression of the signature genes.

## Drug sensitivity analyses

The pRRophetic [36] package was utilized to perform half-maximal inhibitory concentration (IC50) analyses to predict drug sensitivity in different risk groups.

## Generation and analysis of single-cell transcriptomes

Using default parameters, the raw reads were demultiplexed and mapped to the reference genome via the 10×Genomics Cell Ranger pipeline (https://support.10xgenomics.com/single-cell-geneexpress/software/pipelines/latest/what-is-cell-ranger). All downstream single-cell analyses were conducted through Cell Ranger and Seurat. For each gene and each cell barcode (filtered by Cell Ranger), unique molecular identifiers (UMIs) were calculated to construct a digital expression matrix. Seurat filtering criteria were applied: mitochondrial gene proportion < 15%, genes expressed in more than three cells were considered expressed, and each cell was required to have at least 200 expressed genes. After filtering out some extraneous cells, Cell Ranger Count was used to perform alignment, filtering, barcode counting, and UMI counting from the FASTQ files. It utilized Chromium cell barcodes to generate feature-barcode matrices through either Cell Ranger Count or Cell Ranger Aggr and reran dimensionality reduction, clustering, and gene expression algorithms with default Cell Ranger settings.

Next, Seurat was employed for secondary analysis of gene expression. Specifically, the "Seurat" package was used for data normalization, dimensionality reduction, clustering, and differential expression analysis. For integrated analysis of datasets, we applied canonical correlation analysis (CCA) using Seurat's alignment method. For clustering, highly variable genes were selected, and graphs were constructed based on the principal components of these genes, with a resolution of 0.5 for segmentation. The "CellChat" package was utilized for cell communication analysis.

## Cell Markers

B cells (CD79A and MS4A1), Plasma cells (SDC1, MZB1, and IGHG1), NK cells (FGFBP2, KLRD1, and NKG7), CD8$^+$ T cells (CD3E and CD8A), Treg cells (FOXP3, IL2RA, and CTLA4), Endothelial cells (PECAM1 and VWF), Cycling cells (MKI67 and TOP2A), Monocytes (S100A9 and S100A8), M2 Macrophages (C1QA, C1QB, and MRC1), M0 Macrophages (C1QA and C1QB), Epithelial or tumor cells (EPCAM, ALDH1A1, and ALB), Fibroblasts (ACTA2 and COL1A2), DC cells (CD1C and CD1E).

## Quantitative Real-time PCR (qPCR)

Total RNA samples were extracted from treated cells using TRIzol (Invitrogen, USA). 1000 ng of total RNA was reverse transcribed into cDNA for qPCR with Evo M-MLV RT Premix (Accurate Biology, China). Amplification was performed using the SYBR Green premix Pro Taq HS qPCR Kit (Accurate Biology, China), and mRNA expression levels were analyzed with the LightCycler 96 instrument (Roche, Switzerland). The relative mRNA expression levels of target genes were determined by normalizing to glyceraldehyde-3-phosphate dehydrogenase (GAPDH) mRNA levels. Data was analyzed using the $2^{-\Delta\Delta CT}$ method. Primers used were synthesized by TIANYI HUIYUAN (Beijing, China). For GAPDH, the forward primer is '5'-GGACCTGACCTGCCGTCTAG-3" and the reverse primer is '5'-GTAGCCCAGGATGGCCTTGA-3." DNTTIP2 uses '5'-CAAGTGAGGTTGCCATTGAGG-3" as the forward primer and '5'-GTCTATGCTGCTGCTGTGCTTCAACT3-" as the reverse primer.

## Western blotting

72 hours after cell transfection, cells were lysed at 4°C using RIPA buffer containing PMSF. After 30 minutes, the lysates were collected into 1.5 mL EP tubes. The cells were further lysed by sonication (5 seconds on, 5 seconds off) for a total of 3 minutes. The lysates were then centrifuged at 12,000 rpm for 10 minutes at 4°C, and the supernatant was transferred

to new EP tubes. Protein concentrations were determined using a BCA protein assay kit (Beyotime Biotechnology), and equalized across groups. 6×loading buffer was added at the appropriate ratio, and the samples were heated in a metal bath at 100°C for 10 minutes before being stored at −80°C for further use. Protein samples were taken out from the −80°C freezer and heated in a metal bath at 100°C for 10 minutes. After a brief centrifugation and mixing, the samples were loaded into the wells of an SDS-PAGE gel. Electrophoresis was performed at a constant voltage, starting at 80V and then switching to 120V. The PVDF membrane was soaked in methanol for 10 minutes and transferred using wet transfer at 300 mA in an ice bath. After transfer, the membrane was blocked with 5% skim milk at 37°C on a shaker for 2 hours. The membranes were incubated with the following primary antibodies overnight at 4°C: anti-β-tubulin (Cell Signaling Technology, β-Tubulin (9F3) Rabbit mAb #2128, 1:1000), anti-GAPDH (Cell Signaling Technology, GAPDH (D16H11) Rabbit mAb #5174, 1:1000), anti-HBx (Abcam, Recombinant Anti-Hepatitis B Virus protein X antibody [EPR27041–49] (ab309352), 1:1000), anti-DNTTIP2 (ThermoFisher Scientific, TDIF2 Polyclonal Antibody, PA5–60731, 1:1000), anti-MIF (Cell Signaling Technology, MIF (E7T1W) Rabbit mAb #87501, 1:1000) and anti-CD74 (Servicebio, CD74 Rabbit pAb #GB115427, 1:1000). The next day, the membrane was washed five times with TBST and once with TBS, each for 5 minutes. It was then incubated with secondary antibody on a shaker at room temperature for 45 minutes [Horseradish peroxidase (HRP)-conjugated anti-rabbit IgG antibody (Zhongshan Golden Bridge Biotechnology, 1:5000)]. Then the membrane was washed five times with TBST and once with TBS, each for 5 minutes. Chemiluminescence detection was performed using SuperSignal West Femto Maximum Sensitivity Substrate (Thermo, USA).

### Cell counting Kit-8 (CCK-8) cell viability assay

Cells transfected with either NC plasmid or HBx plasmid were seeded into 96-well plates at a density of 8000 cells per well, with 100 μL of complete medium. After 12 hours of incubation, the original culture medium was discarded, and fresh complete medium containing different concentrations of sorafenib, lapatinib, gefitinib, and erlotinib (corresponding to the IC50 of each drug) was added. The cells were then incubated for 24 hours. Subsequently, 10 μL of CCK-8 reagent was added to each well, and cell viability was measured by detecting the absorbance at 450 nm using a microplate reader.

### HCC tissue sample

This study was approved by the Ethics Committee of the Eighth Affiliated Hospital of Sun Yat-sen University (NO. 2024-233-01) and conducted by the recommendations of the International Committee of Medical Journal Editors. Between January 2020 and August 2024, HCC tissues from 3 HBV-infected HCC patients and their corresponding adjacent non-tumor tissues, as well as HCC tissues from 3 HCC patients without HBV infection and their corresponding adjacent non-tumor tissues, were collected at the Eighth Affiliated Hospital of Sun Yat-sen University. All patients were histologically confirmed as stage III HCC, and the HBV-infected patients were diagnosed with chronic hepatitis B.

### Histological analysis

Hematoxylin and eosin (H&E) stained sections were microscopically re-evaluated to confirm the histological diagnosis of each sample. Paraffin-embedded tissue sections were dewaxed and rehydrated using xylene and a series of ethanol solutions with varying concentrations. After deparaffinization, sections were stained with hematoxylin for 5 minutes, rinsed in ddH$_2$O, and then stained with eosin for 5 minutes. This was followed by dehydration in ethanol, clearing in xylene, and mounting with neutral balsam. Immunohistochemical (IHC) analysis was performed using the above tissue samples. After deparaffinization, antigen retrieval was carried out using Tris-EDTA buffer (Servicebio, pH 9.0). The sections were permeabilized with 0.1% Triton X-100 and blocked with 5% BSA to prevent nonspecific binding. Primary antibody was then applied, and the slides were incubated overnight at 4°C: anti-DNTTIP2 (ThermoFisher Scientific, TDIF2 Polyclonal Antibody, PA5–60731, 1:500), anti-MIF (Cell Signaling Technology, MIF (E7T1W) Rabbit mAb #87501, 1:400) and anti-CD74 (Servicebio, CD74 Rabbit pAb #GB115427, 1:1000). After washing with PBS, a labeled secondary antibody was added

and incubated at room temperature for 60 minutes: Goat anti-Rabbit IgG-HRP Antibody (Absin, 1:500). Following another round of washing, DAB substrate (Servicebio) was added for color development, and the staining was monitored under a microscope to determine the optimal stopping time. The sections were then counterstained with hematoxylin (Servicebio) for 2 minutes and rinsed with running water. Finally, the slides were dehydrated through graded ethanol, cleared in xylene until transparent, mounted with neutral balsam, and observed under a light microscope.

## Statistical analysis

One-way ANOVA and Kruskal-Wallis tests were used to assess the significance of differences among three or more groups. The Wilcoxon test was applied for comparisons between two groups. Correlation analyses were based on Spearman and distance correlation methods. The univariate Cox regression model was used to calculate the hazard ratio (HR) and 95% confidence interval (CI). The multivariate Cox regression model was applied to assess the value of the risk score as an independent prognostic biomarker for patient prognosis. All data processing was performed using R 4.1.0 software. GraphPad Prism 10.0 software (La Jolla, CA, USA) was used for analysis. All experiments were repeated three times.

## Results

### Differential gene expression analysis and functional annotation in HCC cells expressing HBx

The workflow of this research is illustrated above (Fig 1). HepG2 cells were generated by transfecting with HBx or control plasmids respectively. Sample information and RNA sequencing data were uploaded to the Gene Expression Omnibus (GEO) database (GSE276530). Based on three algorithms, a differential analysis was conducted on HepG2 cells expressing HBx compared to control group ($|\log_2FC|>0.5$, P<0.05). The DESeq2 algorithm identified 377 downregulated genes and 256 upregulated genes (Fig 2A), edgeR found 491 downregulated and 335 upregulated genes (Fig 2B), and limma detected 363 downregulated genes and 353 upregulated genes (Fig 2C). Further intersecting the results, 279 downregulated genes and 234 upregulated genes were obtained (Fig 2D). Subsequently, Gene Ontology (GO) and Kyoto Encyclopedia of Genes and Genomes (KEGG) annotation and pathway enrichment analyses were conducted for the 513 differentially expressed genes. The results of the GO enrichment functional annotation indicated that differentially expressed

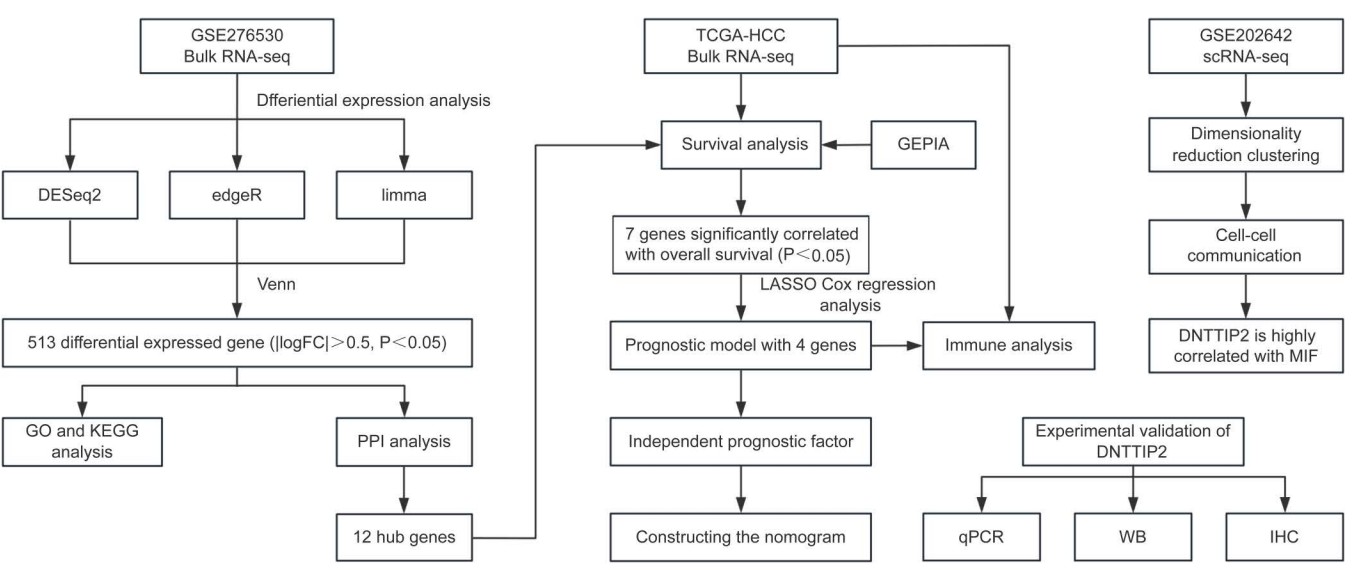

**Fig 1. Workflow chart of the study.**

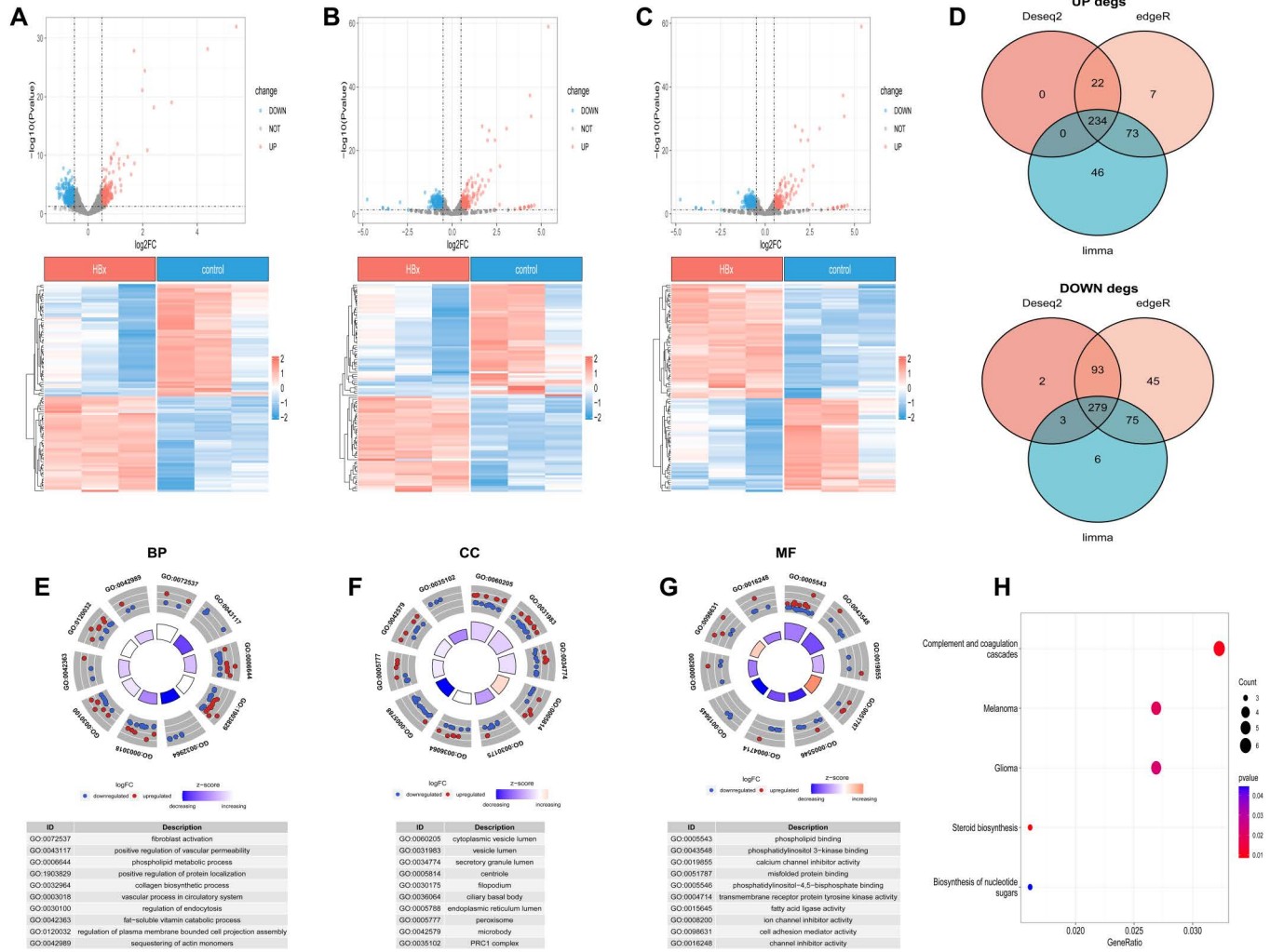

**Fig 2. DEGs and enrichment analysis in HepG2 cells expressing HBx.** A-C Volcano plot and heatmap of DEGs in (A) Deseq2, (B) edgeR, (C) limma. D Venn diagrams showed the number of upregulated and downregulated. E-G The enriched GO terms of DEGs: (E) Biological Process, (F) Cellular Component, and (G) Molecular Function. Red and blue dots represent upregulated and downregulated genes based on $\log_2$FC. Inner box color indicates pathway z-score, and box size reflects enrichment significance. H KEGG pathway enrichment results of DEGs.

genes (DEGs) were primarily enriched in biological processes associated with fibroblast activation, blood vessel formation, matrix production, endocytosis regulation, and metabolism (Fig 2E). Regarding cellular components, DEGs were predominantly enriched in vesicles, microsomes, peroxisomes, and the endoplasmic reticulum lumen (Fig 2F). In terms of molecular functions, they were primarily enriched in pathways related to phospholipid metabolism, fatty acid metabolism, and ion channels (Fig 2G). The KEGG functional enrichment results demonstrated that the DEGs were mainly enriched in pathways such as the complement and coagulation cascades, steroid biosynthesis, and ribosome biogenesis (Fig 2H).

### Identification of prognostic hub genes

Protein interaction data were obtained from the String database. The interaction information for the 513 DEGs was visualized using Cytoscape (S1A Fig). The subnetwork module with the highest score (Score: 11.273) was identified

using MCODE. This module comprises 12 genes (Fig 3A). Survival analysis was conducted on the TCGA-HCC cohort to identify prognosis-related genes. The results indicated that patients with high expression of seven genes (NVL, WDR75, NOP58, BRIX1, DNTTIP2, NIFK, and RPF2) exhibited poor prognoses (Fig 3B). Furthermore, survival analysis performed using HCC data integrated into the GEPIA database corroborated these findings (Fig 3C). The expression levels of the seven prognosis-related genes were significantly elevated in tumor samples compared to paired non-tumor samples (Fig 3D). Spearman correlation analysis demonstrated a positive correlation among the expression levels of these seven prognostic-related genes (Fig 3E). Our results revealed that the expression levels of these seven genes were significantly upregulated in HCC cells expressing HBx and correlated with poor prognosis in HCC patients. Consequently, these seven genes were identified as prognostic hub DEGs.

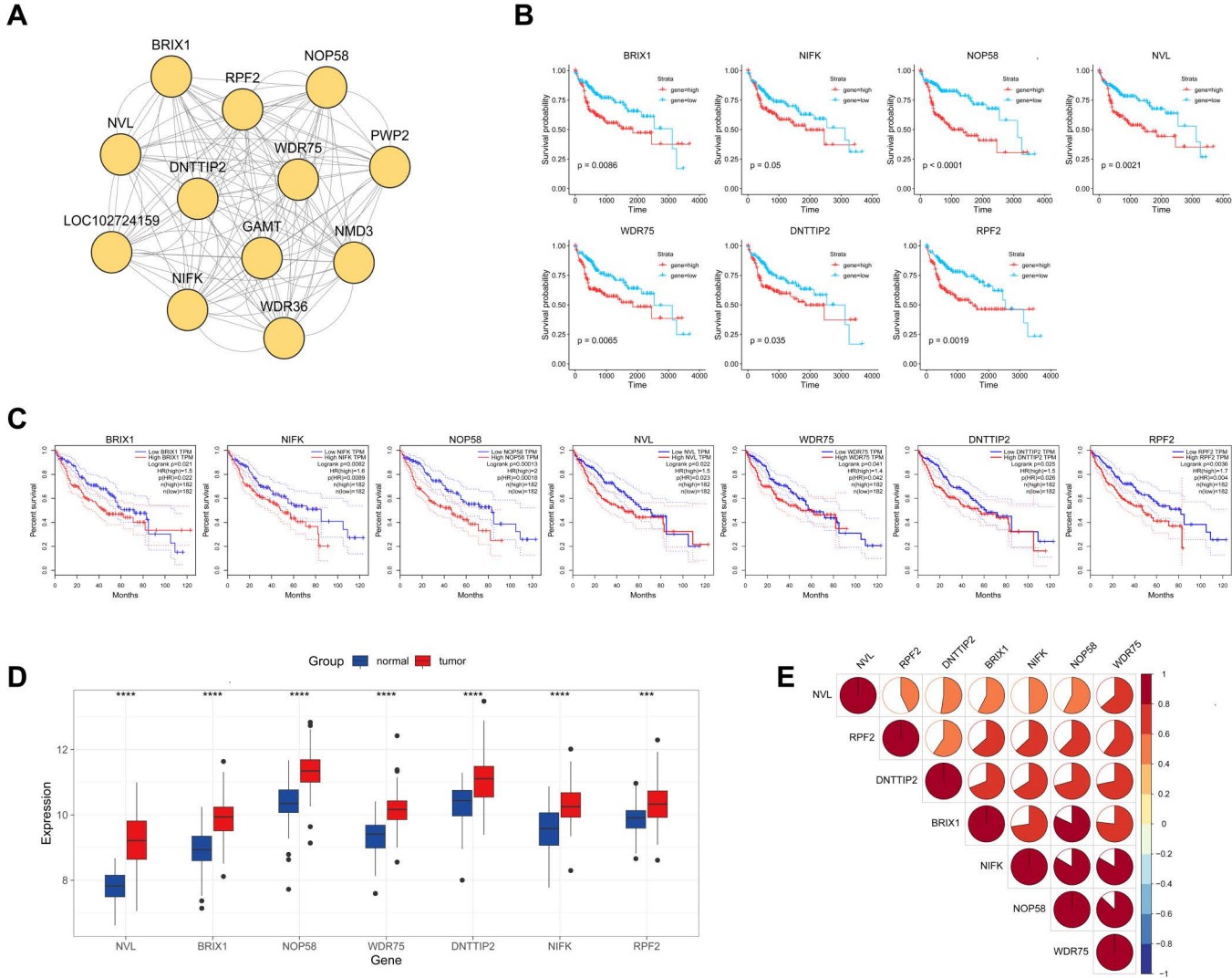

**Fig 3. PPI Network Analysis, Hub DEGs Identification, and Survival Analysis in HCC.** A A key cluster with 12 genes by MCODE based on the PPI network. B KM-plot of prognostic HBx-related hub genes based on TCGA cohort. C KM-plot of prognostic HBx-related hub genes based on GEPIA. D The expression of prognostic hub DEGs in the normal and tumor tissues (***P < 0.001, (****P < 0.001). E The correlation between the seven prognostic hub DEGs using Spearman analyses. Positive correlation was marked with red. Colors represent different ranges of Pearson correlation coefficients (e.g., dark red indicates R = 0.8 - 1.0). The colored areas are categorized based on correlation strength, not proportionally scaled to the R value.

## Construction of prognostic riskscore signature

Among the seven prognosis-related genes, four genes with non-zero coefficients were retained by LASSO regression for constructing the riskscore model, while the remaining three were excluded due to minimal predictive contribution. The risk-score signature was developed considering the hub genes' significant role in patient prognosis. The riskscore was calculated using LASSO Cox regression from the expression levels of four hub genes: BRIX1, RPF2, DNTTIP2, and WDR75 (Fig 4A, 4B). Patients were categorized into the high-risk and low-risk groups based on the cutoff value of 5.39294, which was determined using the MaxStat R software package (Fig 4C). The low-risk group had a significant survival benefit (Fig 4D). The mortality rate increased significantly with a higher riskscore (Fig 4E). Furthermore, the expression levels of the hub genes in the high-risk group were notably higher than those in the low-risk group (Fig 4F). Multivariate Cox regression

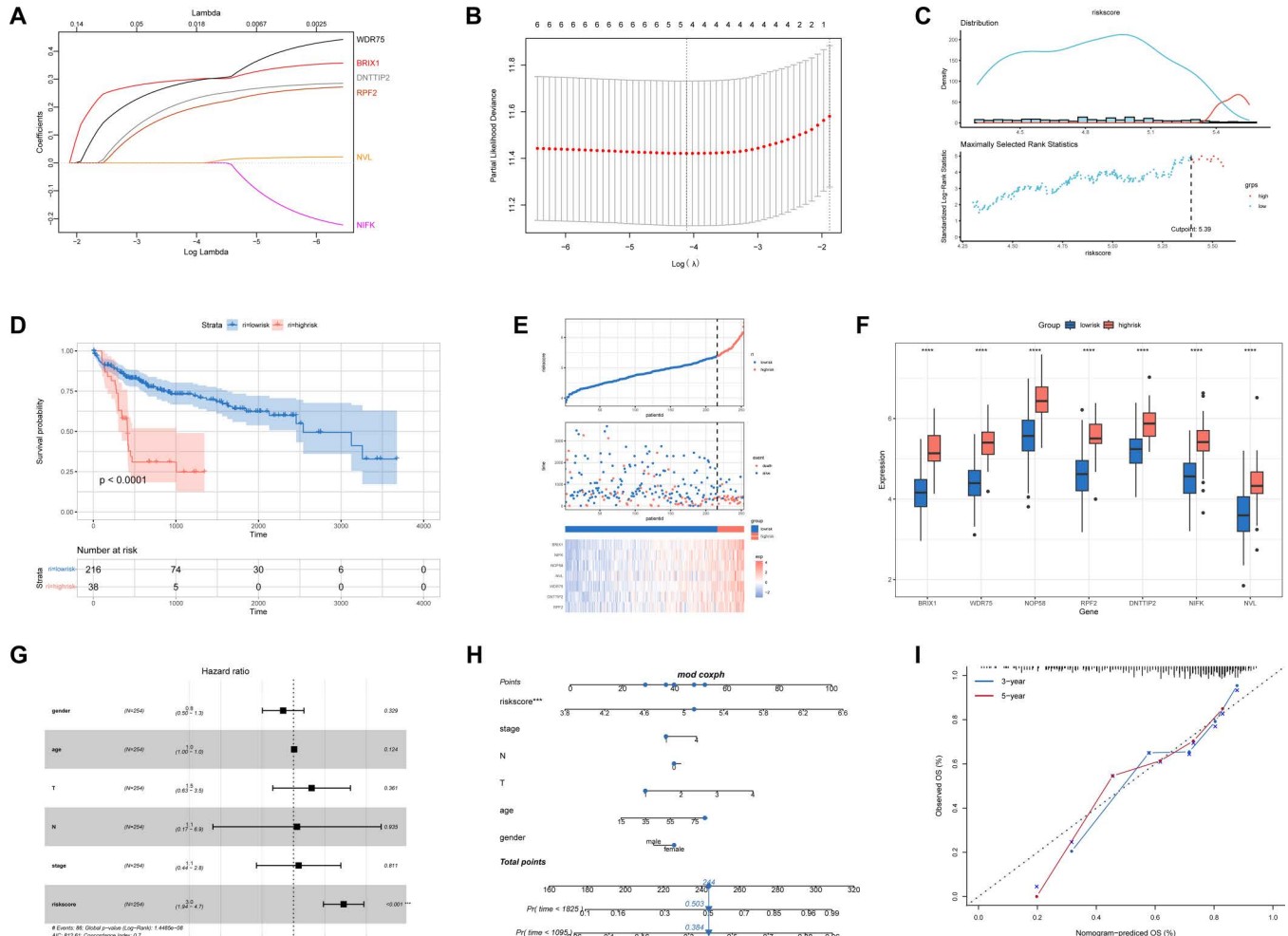

**Fig 4. Construction and prognostic value of the riskscore signature.** A Selection of the optimal parameter (lambda) in the LASSO model. B LASSO coefficients of the 7 hub DEGs in TCGA cohort. C The optimal cut-off point to divide riskscore into low and high groups was 5.39294. The height of each column in the histogram represents the number of patients within the corresponding riskscore range. D Overall survival analysis for the low-risk and high-risk groups in TCGA cohort. E The mortality risk in the low-risk and high-risk group patients in the TCGA cohort. F The hub DEGs expressed in the low-risk and high-risk groups (****P < 0.0001). G Multivariate Cox analysis for the clinicopathologic characteristics and riskscore in TCGA cohort. H A nomogram to predict the prognostic of HCC patients. I Calibration plots showing the probability of 3-, and 5-year overall survival in TCGA cohort. The 45-degree line represented the ideal prediction.

analysis, which included variables such as age, gender, clinical stage, T stage, and N stage, indicated that the riskscore signature served as an independent prognostic biomarker for assessing the prognosis of HCC patients (Fig 4G). A nomogram was created by integrating the riskscore with independent clinical prognostic factors (Fig 4H), which was designed to predict the probability of mortality in patients with HCC. The calibration diagram demonstrated that the prediction model's performance was commendable compared to the ideal model (Fig 4I), thereby providing an effective method for predicting the prognosis of HCC patients.

## Immune cell infiltration and potential therapeutic targets

Infiltrating immune cells are crucial in regulating tumor progression within the TME and are closely associated with prognosis [37]. This study further examined the roles of hub genes, including BRIX1, RPF2, DNTTIP2, and WDR75, in TME immune cell infiltration. An evaluation of the TCGA-HCC dataset revealed distinct differences in the infiltration of various immune cells between normal and tumor samples. Specifically, activated CD4+T cells and CD56dim NK cells were enriched in tumor tissues. In contrast, other subtypes, such as monocytes, macrophages, CD8+T cells, NK cells, and Th1 cells, were predominantly found in normal tissues (Fig 5A). Furthermore, the expression of hub genes demonstrated a negative correlation with anti-tumor cells, including NK cells, Th1 cells, and effector memory CD8+T cells. At the same time, it exhibited a positive correlation with cells that may promote tumor progression, such as activated CD4+T cells, dendritic cells, NK T cells, central memory CD4+T cells, effector memeory CD4+T cells and Th2 cells (Fig 5B). The ESTIMATE algorithm indicated reduced immune and stromal activity in HCC tissues (Fig 5C, 5D). The results also highlighted connections between hub genes and specific treatment targets. Notably, WDR75 showed the strongest association with rapidly accelerated fibrosarcoma (RAF) (Fig 5E), while DNTTIP2 and BRIX1 were most closely related to CTLA4 (Fig 5F, 5G). Additionally, RPF2 exhibited the highest correlation with vascular endothelial growth factor receptor (VEGFR) (Fig 5H). These findings may provide valuable insights for enhancing the treatment of HCC patients.

## Tumor immune landscape and drug sensitivity analysis in high- and low-risk groups

The heterogeneity of the TME significantly influences tumor progression and prognosis. Immune infiltration analysis revealed that the high-risk group exhibited elevated levels of Th2 cells accompanied by reduced levels of Th1 cells and NK cells (Fig 6A). The correlation between the riskscore and immune infiltrating cells was illustrated (Fig 6B), showing varying degrees of association. It further illustrates the relationships between the risk score and key genes involved in immunotherapy and targeted therapy (Fig 6C). Differences between the Tumor Immune Dysfunction and Exclusion (TIDE) [34] and the immunopheno score (IPS) [35] were analyzed to evaluate the immune landscape of both the high-risk and low-risk groups. The results indicated that TIDE, immune exclusion, and the myeloid-derived suppressor cell (MDSC) scores were significantly higher in the high-risk group compared to the low-risk group, suggesting a greater possibility of immune escape in the former (Fig 6D-6G). The IPS scores for CTLA(+)PD1(-), CTLA(-)PD1(+), CTLA(+)PD1(+), and CTLA(-)PD1(-) were all significantly lower in the high-risk group (Fig 6H-6K). These findings suggest that patients in the high-risk group may exhibit lower tumor immunogenicity and reduced responsiveness to immune checkpoint inhibitors, including anti-PD1 and anti-CTLA4 therapies. These findings are further supported by the significant differences in cohort distributions regarding patients' responses to immunotherapy between the high-risk and low-risk groups (Fig 6L). Additionally, a drug sensitivity analysis was conducted to identify potential therapeutic agents. The pRRophetic algorithm was employed to predict the sensitivity of patients in both risk groups to four commonly used anticancer drugs: sorafenib, lapatinib, erlotinib, and gefitinib. The results demonstrated that patients in the low-risk group exhibited greater sensitivity to sorafenib, whereas those in the high-risk group were more responsive to lapatinib, erlotinib, and gefitinib (Fig 6M), which was further supported by CCK-8 assay results (Fig 6N). These findings suggest that the riskscore can be an essential indicator for tailoring treatment strategies to individual patients.

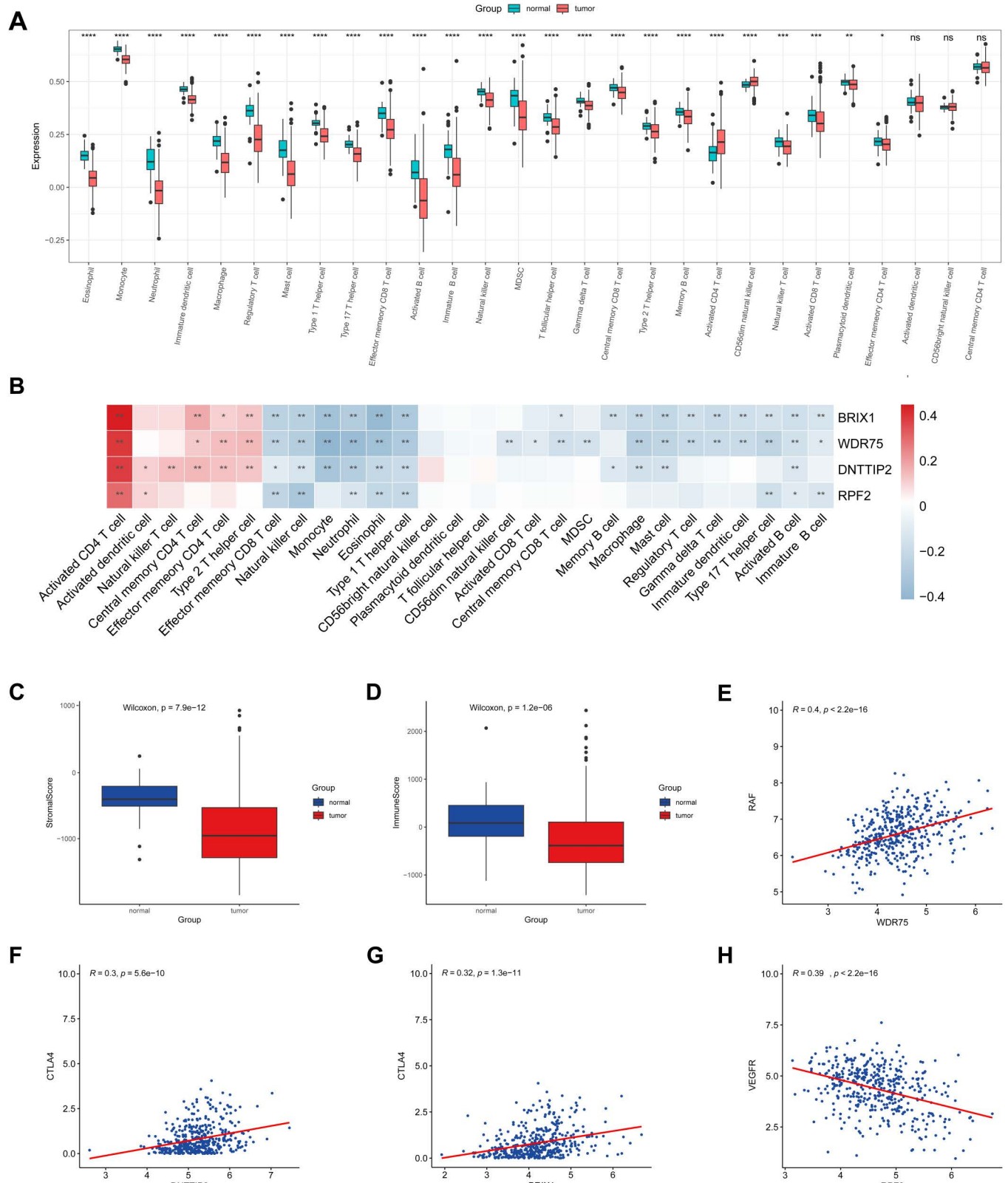

**Fig 5. TME immune cell infiltration and correlation between LASSO and therapeutic target genes.** A Enrichment scores of TME infiltration cells in normal liver versus HCC tissues. Higher scores represent higher levels of immune cell infiltration. B The correlation between LASSO genes and

TME infiltration cells. Red, positive; Blue, negative. C The boxplot of stromal score. D The boxplot of immune score. E The correlation between WDR75 expression and RAF expression. F The correlation between DNTTIP2 expression and CTLA4 expression. G The correlation between BRIX1 expression and CTLA4 expression. H The correlation between RPF2 expression and VEGFR expression. (*P<0.05; **P<0.01; ***P<0.001; ****P<0.0001).

### Inference of cell-cell communication in TME

A deeper understanding of the complexities of TME infiltration is essential. ScRNA-seq from HBV-related HCC (GSE202642) was obtained from the GEO database. The cells were classified into 13 distinct cell types based on defined marker genes (Fig 7A). CellChat was employed to infer complex signaling in cell-cell communication, illustrating the strength of incoming and outgoing interactions between different cell types (Fig 7B, S1B). Notably, M2 macrophages emerged as the predominant population expressing receptors and ligands involved in these interactions, highlighting their central role in TME communication. Additionally, the MIF signaling pathway exhibited the highest intensity of action within malignant cells (Fig 7C, S1D). Immunosuppressive-related cells, including M2 macrophages and Treg cells, were identified as receptors for the MIF signaling (Fig 7D). The MIF-(CD74+CXCR4) ligand-receptor pair significantly contributed to the MIF signaling pathway (Fig 7E). These findings suggest that malignant cells in HBV-related HCC may regulate the activities of M2 macrophages and Treg cells via the MIF signaling pathway, resulting in the emergence of an immunosuppressive TME.

### DNTTIP2 association with MIF pathway in HCC immunosuppression

The analysis indicated that the MIF signaling pathway was crucial to the development of the TME in HBV-related HCC. ScRNA-seq analysis revealed that DNTTIP2 displayed a high expression level in malignant cells, and we also observed elevated MIF expression in the same cellular context (Fig 8A). Correlation analysis demonstrated that MIF expression has the strongest association with DNTTIP2 expression among key molecules (Fig 8B). As a critical molecule, these results suggest that DNTTIP2 may modulate the TME associated with the MIF signaling pathway to induce immunosuppression. Additionally, relevant experiments were performed to confirm DNTTIP2 expression. The result of qPCR and Western blot analysis indicated that DNTTIP2 mRNA and protein expression was significantly upregulated in HepG2 cells expressing HBx (Fig 8C, 8D). Moreover, we examined the expression of MIF and CD74, key molecules in the MIF signaling pathway, and found that their protein levels were also elevated in HBx-overexpressing HepG2 cells (Fig 8D). Furthermore, HCC tissue samples were collected from both HBV-positive and HBV-negative patients, and H&E staining was used to confirm the samples as HCC tissues (Fig 8E). DNTTIP2, MIF and CD74 expression levels in HBV-positive HCC tissues, marked by brown staining, exceeded those in HBV-negative HCC tissues (Fig 8E). These findings suggest that the MIF signaling pathway may be a key pathway leading to the formation of an immunosuppressive TME, with DNTTIP2 as a potential key molecule.

## Discussion

HCC poses a significant threat to human health, with over half of HCC cases linked to HBV infection. Many HCC patients present with advanced-stage tumors, which limit their treatment options to systemic therapies, including immunotherapy and targeted therapies [28]. While various immunotherapeutic strategies have been implemented for HCC, early diagnosis, treatment, and identifying more effective immunotherapeutic approaches remain major challenges [38]. The composition of TME significantly influences patient responsiveness to immunotherapy [25]. A deeper understanding of TME heterogeneity can facilitate the selection of more appropriate treatments. HBx, a multifunctional protein produced by HBV, promotes HCC through multiple mechanisms. However, the specific ways in which HBx regulates the TME remain unclear [17]. Further investigation may uncover new therapeutic targets and innovative strategies for HCC management.

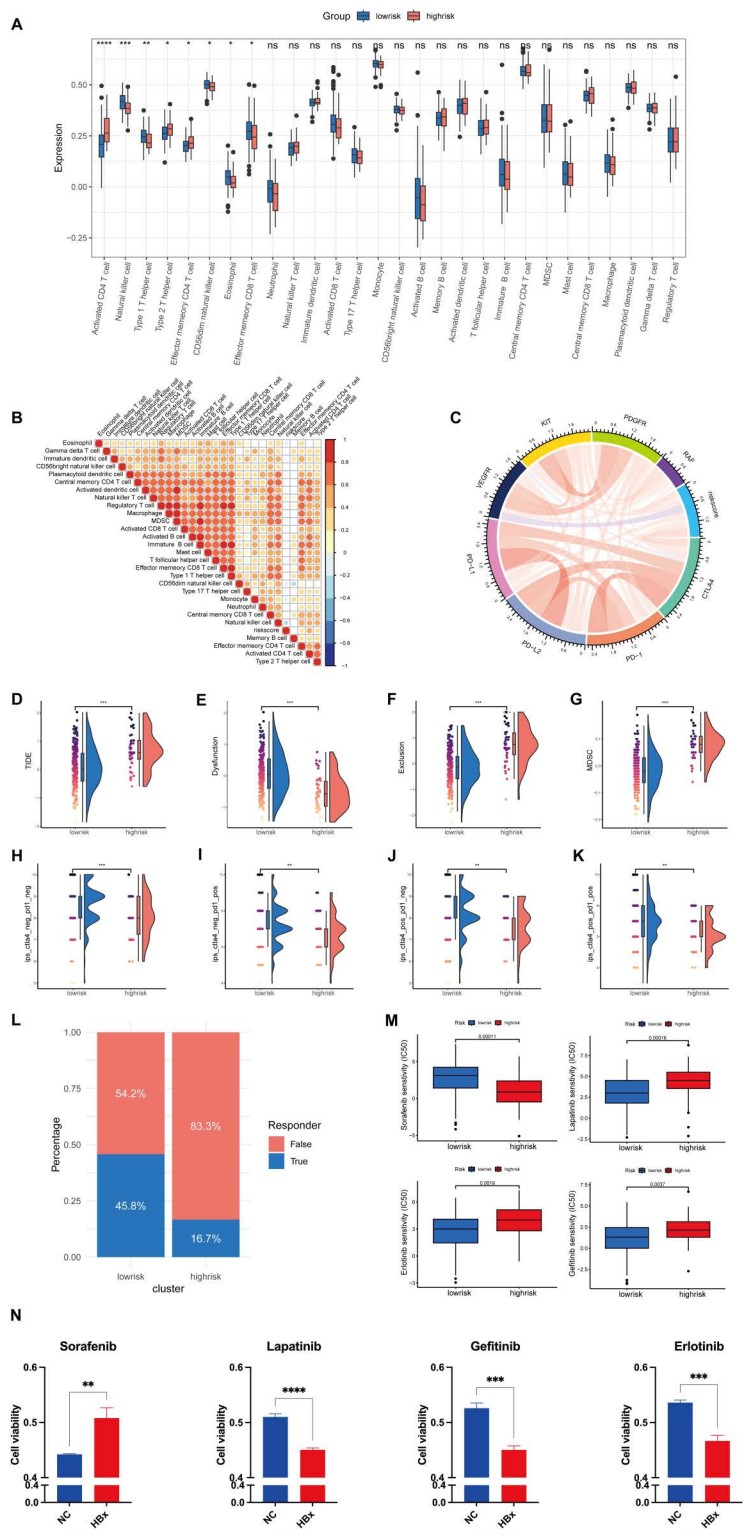

**Fig 6. Immune landscape and drug sensitivity in high-risk and low-risk groups.** A Differences in TME infiltration cells between high-risk and low-risk groups. Enrichment scores represent the relative abundance of immune cell infiltration; higher scores indicate greater infiltration levels. B The correlation between riskScore and TME immune cell types. The size of each dot corresponds to the absolute risk score (|R|), with larger dots indicating

stronger correlations. C The correlation between riskScore and genes targeted by immunotherapy or targeted therapy. The width of the chords represents the strength of the correlation, with wider connections indicating higher |R| values. D-G Alteration of (D) TIDE score, (E) immune dysfunction, (F) immune exclusion and (G) MDSC in high-risk and low-risk groups. H-K IPS scores of PD1 and CTLA4 in high-risk and low-risk groups. Low IPS scores in high-risk tumors suggest poor response to immune checkpoint inhibitors, while higher IPS scores in low-risk tumors suggest a better response. (H) Negative response to PD1 treatment and CTLA4 treatment [CTLA(-)PD1(-)]. (I) Positive response to PD1 treatment and negative response to CTLA4 treatment [CTLA(-)PD1(+)]. (J) Negative response to PD1 treatment and positive response to CTLA4 treatment [CTLA(+)PD1(-)]. (K) Positive response to PD1 treatment and CTLA4 treatment [CTLA(+)PD1(+)]. L The cohort distribution of immunotherapy response between high-risk and low-risk groups within the TCGA dataset. M Sensitivity analyses of sorafenib, lapatinib, erlotinib, and gefitinib in high-risk and low-risk groups. N Cell viability of HepG2 cells transfected with control (NC) or HBx plasmid after treatment with sorafenib, lapatinib, gefitinib, and erlotinib. (*P < 0.05; **P < 0.01; ***P < 0.001; ****P < 0.0001).

Previous studies have developed HCC prognostic models based on cuproptosis, mitochondrial unfolded protein responses, and other gene signatures [39,40]. However, no prior investigations have specifically focused on HBx-related genes. Our study addressed this gap by identifying DEGs associated with HBx-expressing HCC and establishing a novel prognostic model incorporating four key HBx-related hub genes. Unlike previous approaches, our model highlights the specific impact of HBx on HCC prognosis and tumor immunity, offering new insights into the unique role of HBx in shaping the tumor microenvironment. Although the TCGA-LIHC dataset used for model construction does not contain information on HBV infection status – making it unclear whether all 374 HCC samples were from HBV-infected patients – the integration of HBx-associated genes ensures the model's relevance to HBV-related HCC. This model not only stratifies patients into low- and high-risk groups with distinct survival outcomes but also predicts responses to immunotherapy, demonstrating its potential clinical utility.

The identified prognostic model consists of four genes, all serving as risk factors affecting prognosis. Biogenesis of ribosomes in xenopus laevis 1 (BRIX1) is essential for 60S ribosomal subunit biosynthesis and is implicated in cancer progression [41]. It promotes glycolysis and cell proliferation in colorectal cancer and acts as a pro-cancer factor in gastric cancer, with potential as a biomarker and therapeutic target [42]. Its expression correlates with poor prognosis in HCC [43], supporting our findings. WD repeat domain 75 (WDR75), integral to ribosome biosynthesis, is required for pre-rRNA transcription. The depletion of WDR75 impairs proliferation and induces cellular senescence by activating the RPL5/RPL11-dependent p53 stabilization checkpoint [44]. Currently, research on WDR75 in tumors is limited, necessitating further investigation. Ribosome production factor 2 (RPF2) participates in the assembly of the large ribosomal subunit and may regulate the localization of 5S RNP/5S [45]. In colorectal cancer, RPF2 is significantly overexpressed, promoting epithelial-mesenchymal transition (EMT) via the AKT/GSK-3β signaling pathway, thereby enhancing migration and invasion of colorectal cancer cells in vitro and vivo [46]. Moreover, a recent study has indicated that RPF2 was highly expressed in liver cancer tissues compared to adjacent normal tissues. RPF2 overexpression facilitates cell proliferation, migration, and invasion, contributing to the progression of liver cancer [47]. Interestingly, in our study, we observed a negative correlation between RPF2 expression and VEGFR expression. While this may seem counterintuitive, especially considering RPF2's role in promoting liver cancer progression, this correlation does not imply a direct causal relationship. It rather suggests that RPF2 may promote tumor progression through alternative pathways independent of VEGFR, highlighting the potential for targeting different pathways in HCC treatment.

DNTTIP2, as the most highly expressed and significant gene among them, regulates the transcriptional activity of DNA Nucleotidylexotransferase (DNTT) and Estrogen Receptor 1 (ESR1), likely functioning as a chromatin remodeling protein [48,49]. DNTTIP2, also known as terminal deoxynucleotidyltransferase interacting factor 2 (TdIF2), plays key roles in both cellular processes and tumorigenesis. It is an acidic protein that interacts with TdT and has been identified as a histone chaperone within the nucleus. It binds to DNA and core histones, and its C-terminal region is rich in acidic amino acids, which are important for its function in chromatin dynamics [50]. In the context of cancer, DNTTIP2 has been shown to have a significant role in regulating the cell cycle, particularly in pancreatic cancer

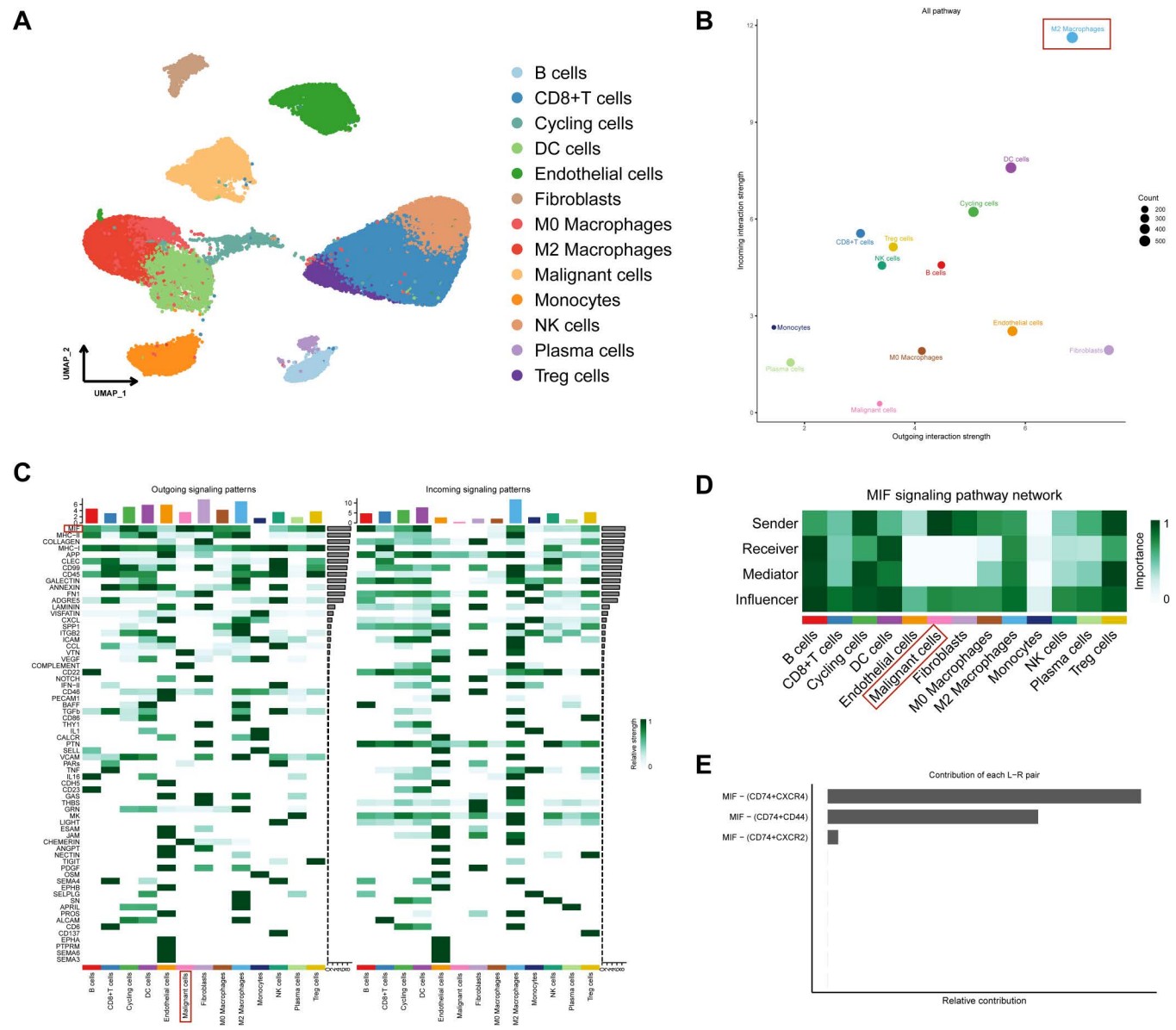

**Fig 7. Identification of HBV-related HCC cell subtypes and inference of cell-cell communications in TME.** A The UMAP plot shows different cell types in HBV-related HCC tissues. B Illustration of the incoming and outgoing interaction strengths for each cell type. C The incoming and outgoing signaling pathways of each cell type. D The heatmap shows the communication probability of the MIF signaling pathway. E Contribution of each ligand-receptor(L-R) pair in the MIF signaling pathway.

(PDAC). Depletion of DNTTIP2 leads to G1 arrest in MIA-PaCa-2 cells and G2 arrest in PK-1 cells, with a corresponding downregulation of cell cycle regulators such as SATB1, CDK6, and CDK1. However, DNTTIP2 depletion does not induce apoptosis, highlighting its more direct role in cell cycle regulation rather than in promoting cell death [51]. Additionally, DNTTIP2 has been shown to influence ribosomal RNA (rRNA) gene transcription by interacting with human ribosomal RNA promoters and promoting hrDNAP activity. Through its interaction with the histone acetyltransferase Tip60, DNTTIP2 inhibits Tip60's acetylation activity and co-localizes with Tip60 in the nucleolus. This

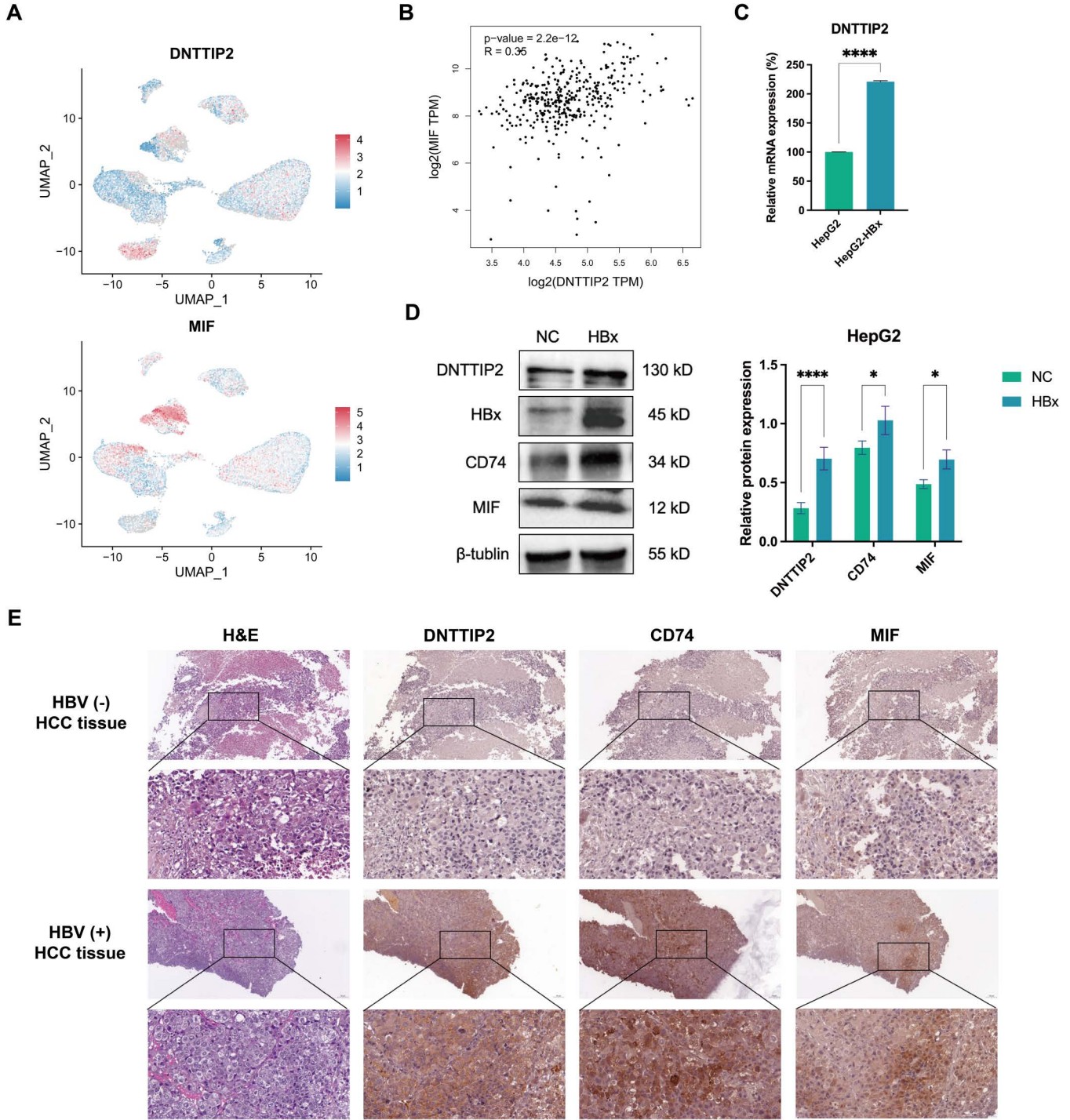

**Fig 8. Correlation of DNTTIP2 with MIF and verification of DNTTIP2, MIF and CD74 expression in HBV-related hepatocellular carcinoma.** A Gene expression levels of DNTTIP2 and MIF in GSE202642 dataset. B Correlation between DNTTIP2 and MIF expression based on TCGA cohort. C Expression of DNTTIP2 mRNA in HepG2 cells transfected with HBx plasmid or control plasmid (NC). D Expression of DNTTIP2, CD74 and MIF protein after transfection with HBx plasmid and control plasmid (NC) in HepG2 cells. E Representative images of H&E staining and IHC staining of HBV-negative and HBV-positive HCC tissue. IHC staining was performed to detect DNTTIP2, MIF, and CD74 expression. Brown staining indicates positive expression. Scale bars, 100 μm and 20 μm. Data presented as mean±SEM (n=3). *P<0.05; **P<0.01; ***P<0.001; ****P<0.0001.

interaction also promotes acetylation of the upstream binding factor (UBF), a key regulator of rRNA synthesis [50]. DNTTIP2 plays a crucial role in both TME and tumorigenesis. It promotes M2 tumor-associated macrophages (TAMs) activation and angiogenesis, processes that contribute to tumor progression. Additionally, DNTTIP2 is involved in the IL6/JAK/STAT3 signaling pathway, which is key to tumor growth and immune evasion. The amplification of DNTTIP2 is associated with poor prognosis in glioma patients, and its hypermethylation is also linked to unfavorable outcomes, highlighting its potential as a biomarker and therapeutic target [52]. Our results suggest that DNTTIP2 might also be involved in regulating M2 macrophages. Additionally, elevated DNTTIP2 expression has been observed in pediatric acute myeloid leukemia as one of the potential hub genes [53]. These findings imply that DNTTIP2 may correlate with poor prognosis across various tumors. Currently, no studies have reported on the role of DNTTIP2 in HCC. Our study is the first to elucidate the significance of DNTTIP2 as the key HBx-related molecule in predicting the prognosis and treatment of HCC. Furthermore, our findings reveal that DNTTIP2 is strongly correlated with MIF expression and is predominantly expressed in malignant cells. These results suggest that DNTTIP2 may influence the TME by regulating interactions with M2 macrophages and Tregs through the MIF-(CD74 + CXCR4) signaling pathway. This pathway is a known mediator of immunosuppression in various cancers, promoting tumor progression by shaping an immunosuppressive microenvironment. The identification of DNTTIP2 as a key regulator of this process highlights its potential as a prognostic biomarker and a therapeutic target.

Analysis of the immune landscape further supports these findings. High-risk patients, as identified by our model, exhibited diminished infiltration of anti-tumor immune cells, including effector memory CD8 + T cells, NK cells, and Th1 cells, alongside increased infiltration of Th2 cells, which antagonize Th1 cell responses. These patterns suggest that a higher risk score correlates with greater immunosuppression and immune escape. Consistently, high-risk patients demonstrated elevated TIDE, immune exclusion, and MDSC scores, along with lower IPS scores, reflecting a reduced likelihood of responding to immune checkpoint therapy [34,35].

ScRNA-seq provided additional evidence of an immunosuppressive TME in HBV-related HCC. Our results confirmed the presence of immunosuppressive M2 macrophages and Tregs, both associated with poor prognosis [54,55]. Cell-cell communication analysis revealed that M2 macrophages act as central regulators within the TME, receiving signals predominantly from malignant cells via the MIF-(CD74 + CXCR4) ligand-receptor axis [56,57]. This highlights the potential of targeting MIF signaling to disrupt immune evasion mechanisms.

Finally, experimental validation in HBx-expressing HepG2 cells and HBV-infected HCC tissues confirmed elevated DNTTIP2 expression, supporting its role in shaping an immunosuppressive TME. These findings provide novel insights into the interplay between HBx and the TME, underscoring the importance of DNTTIP2 in immune escape. Further research is warranted to elucidate the precise molecular mechanisms underlying these interactions and to explore the therapeutic potential of targeting DNTTIP2 in HCC.

## Conclusion

In conclusion, our study established a novel prognostic model to investigate the significant effects of HBx on the prognosis, immune infiltration, and TME of HCC. This model can effectively predict patient prognosis and immunotherapy responses, facilitating more precise treatment in the clinic. Additionally, this study has identified new targets for future research.

## Supporting information

**S1 Fig. Characterization of HBx-Related DEGs and MIF Signaling Interactions in HBV-Related HCC.** A PPI network of HBx-related DEGs. B The number of interactions and interaction strength between different celltypes in HBV-related HCC tissues. C The interaction strength of MIF signaling pathway between different celltypes in HBV-related HCC tissues.

D Signaling from malignant cells to other cells in the MIF signaling pathway. A thicker line indicates a higher interaction strength or a greater number of interactions.
(PDF)

**S2 Fig. Representative images of H&E staining and IHC staining of DNTTIP2 protein in HBV-negative and HBV-positive adjacent non-tumor tissues.**
(PDF)

**S3 Image. Raw data.**
(ZIP)

## Acknowledgments

The authors acknowledge the Eighth Affiliated Hospital of Sun Yat-sen University for providing the platform.

## Author contributions

**Conceptualization:** Jianhua Zhong, Yuetong Li.

**Data curation:** Ying Jin.

**Formal analysis:** Xinyi Kong.

**Funding acquisition:** Aixia Zhai.

**Investigation:** Miao Qi.

**Methodology:** Yang Liu.

**Project administration:** Changlong Bi, Aixia Zhai.

**Resources:** Yiqi Lin.

**Software:** Jie Qiao.

**Supervision:** Changlong Bi, Aixia Zhai.

**Validation:** Yiling Wu.

**Visualization:** Yaqi Yao.

**Writing – original draft:** Jianhua Zhong, Yuetong Li.

**Writing – review & editing:** Jianhua Zhong, Yuetong Li.

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
