## [Decision Letter · Decision Letter 0]

Dear Dr. Zhai,

Thank you for submitting your manuscript to PLOS ONE. After careful consideration, we feel that it has merit but does not fully meet PLOS ONE’s publication criteria as it currently stands. Therefore, we invite you to submit a revised version of the manuscript that addresses the points raised during the review process.

We look forward to receiving your revised manuscript.

Kind regards,

Li Shen

Academic Editor

PLOS ONE

Journal Requirements:

Reviewers' comments:

Reviewer's Responses to Questions

**Comments to the Author**

1. Is the manuscript technically sound, and do the data support the conclusions?

Reviewer #1: Yes

Reviewer #2: Yes

Reviewer #3: Yes

Reviewer #4: Partly

Reviewer #5: Yes

2. Has the statistical analysis been performed appropriately and rigorously?

Reviewer #1: Yes

Reviewer #2: Yes

Reviewer #3: Yes

Reviewer #4: Yes

Reviewer #5: I Don't Know

3. Have the authors made all data underlying the findings in their manuscript fully available?

Reviewer #1: Yes

Reviewer #2: Yes

Reviewer #3: Yes

Reviewer #4: Yes

Reviewer #5: No

4. Is the manuscript presented in an intelligible fashion and written in standard English?

Reviewer #1: Yes

Reviewer #2: Yes

Reviewer #3: Yes

Reviewer #4: Yes

Reviewer #5: Yes

Reviewer #1: Comments to the Authors:

In this work, Jianhua Zhong and colleagues, used robust bioinformatics analysis and cell experiments to develop a novel HBx-related four-gene prognostic model that may predict clinical outcomes, immune infiltration, and immunotherapy response to HBV-related HCC. Additionally, their findings suggest that DNTTIP2 is a key regulator of immunosuppression that occurs in the TME, driving HCC tumorigenesis and may serve as a putative prognostic biomarker and therapeutic target for HCC.

Below, I have outlined some points for the authors to address:

1. In the discussion section, the authors should present a more robust account of the canonical function of DNTTIP2 and possibly what is already known about its role in the TME and tumorigenesis. Since experiments to elucidate the molecular mechanism(s) by which DNTTIP2 may drive HCC is lacking.

2. The authors stated that “Between January 2020 and August 2024, HCC tissues from 3 HBV-infected HCC patients and their corresponding adjacent non-tumor tissues, as well as HCC tissues from 3 HCC patients without HBV infection and their corresponding adjacent nontumor tissues, were collected at the Eighth Affiliated Hospital of Sun Yat-sen University” but representative images of IHC staining of DNTTIP2 in the adjacent non-tumor tumor tissues are not shown. Could the authors provide that?

Reviewer #2: The present work is interesting and well carried. The information is also well explained in the text, illustrating step by step why the author performs each experiment, and making easier the understanding of the data. The use of bioinformatic algorithms for prediction is very interesting. The conclusions taken by the authors are well based on the data obtained in the experiments and confirmed by different approaches.

However, there are some points that must be clarified:

Page 17, line 208: Although 2^-ddCt is a common technique, it assumes that the amount of sample is doubled at each cycle, and this is not always the case. It would be convenient to use a standard calibration curve using serial dilutions of a pull of samples. It is also important to check more than one normalizer and use GeNorm or Normfinder to find a normalization factor. This reviewer recommends using 2 or 3 in total or at least demonstrating that GAPDH is a good normalizer by showing the Ct's (which should be fairly similar across samples).

Page 19, line 267: Here the threshold is set to |logFC|>0.5 (same for figure 1). In the Methods (page 14, line 118) the values were |log2FC|>1, which makes more sense as it assumes a FC greater than 2-fold. Is the value set here log10? Anyway, there is a mistake that needs to be corrected somewhere.

Page 20, line 283: FOR ALL THE FIGURES’ FOOTNOTES. Knowing that this is not the final format of the paper, this reviewer suggests for the next submission to put all footnotes together, at the end of the text or with the figures or as supplementary material. It is very difficult to find them when you are reading the text and looking at the figures until you realize that they are at the end of each part.

Page 20, line 287: More information is needed for correct interpretation of Figures 2E-G. The outer boxes, with the dots, are divided into four regions or levels where the dots are drawn. What do those levels mean? The inner boxes, where the z-score color is used, have different sizes. Why?

Page 20, line 293: Although the text is fine and I would keep it, Figure 3A is too complicated and, in this reviewer's opinion, does not provide much information (Figure 3B is sufficient to understand the results). I suggest moving Figure 3A to supplementary figures.

Page 21, line 305: The upstream transcription factor analysis is a good idea, but these results seem to be no longer used. Does this figure and result add anything to the paper? This reviewer would remove this figure.

Page 21, line 308: As previously said, I would put all the footnotes together.

Page 21, line 312: More information is needed for correct interpretation of the 3F figures. The color used in the graph is according to the Pearson R value, but is the colored area also proportional to this value?

Page 21, line 314: Since there is no negative correlation in Figure 3F, the blue color comment is not necessary. I would leave just the red color comment.

Page 22, line 334: As previously said, I would put all the footnotes together.

Page 22, line 336: More info is needed for the correct interpretation of the Figure 4C. What do the columns of the histogram represent?

Page 22, line 338: The footnote in Figure 4E said “survival,” and that seems to be the same as Figure 4D, and it is not. It would be better to use the same expression from the main text, “mortality risk”.

Page 23, line 363: As previously said, I would put all the footnotes together.

Page 23, line 364: In Figure 5A, the Y-axis says “expression.” What does that mean? Exactly what is being measured needs to be clarified and explained.

Page 24, line 374: Again, in Figure 6A, the Y-axis said “expression.” What does that mean? Exactly what is being measured needs to be clarified and explained (on page 25, line 395)

Page 24, line 375-376: Figures 6B-C compare not only the risk score to those parameters, but also the parameters to each other (the risk score is only one of the variables). If the comparison between the risk score and the other variables is the important information, the “risk score” should be highlighted in the figure. In the Figure 6B, is the size of the dots proportional to the R score or is it indicated just by the color? In Figure 6C, what does the scale bar on the periphery represent? It would be very useful to have an interpretation of the results as is done on page 24, line 380, 383…

Page 24, line 387: The bioinformatics analysis that the authors have performed to predict sensitivity to anticancer drugs is very interesting, but it would be much more interesting if these results were also verified in a real experiment. This reviewer suggests using cultures of HepG2 cells (with and without HBx plasmid, as the model used by the authors for other experiments) to test the response of these cells to treatment with these drugs. This would increase confidence in the predictions.

Page 25, line 394: As previously said, I would put all the footnotes together.

Page 25, line 395: See Page 24, line 374 comment.

Page 26, line 422: As previously said, I would put all the footnotes together.

Page 27, line 441: “…negative patient, H&E staining…”

Page 27, line 446: As I said above, I would put all the footnotes together. The title of Figure 8 needs to be revised. This reviewer would use “Correlation of DNTTIP2 with MIF and verification of DNTTIP2 expression in HBV-related hepatocellular carcinoma.”

Page 27, line 453: In Figure 8F, since it is sometimes not easy to clearly see the brown color of positive detection, it would be helpful to explain in this text that positivity is marked by that color.

Reviewer #3: The manuscript investigates the immune regulatory role of HBx in hepatocellular carcinoma (HCC) and constructs a prognostic model based on HBx-related genes. Through transcriptome sequencing and database analysis, the authors identified seven HBx-related genes and developed a prognostic risk score model based on four key genes. The study also explores the relationship between these genes and immune cell infiltration in the tumor microenvironment (TME), proposing that DNTTIP2 is highly expressed in HBV-related HCC and is associated with the MIF signaling pathway, potentially playing a key role in the formation of an immunosuppressive TME. The manuscript is well-structured, with a logical flow from experimental design to results and discussion. However, several issues need to be addressed. I suggest a major revision.

1.Result 3.2: Seven genes are identified, but from Result 3.3 onwards, only four genes (BRIX1, RPF2, DNTTIP2, and WDR75) are further analyzed. Please include the remaining three genes in the discussion or provide a clear rationale for excluding them.

2.Result 3.4: The latter half of this section describes the relationship between hub genes and specific treatment targets. This content seems unrelated to the title of Result 3.4. I suggest revising the title to better reflect the content or restructuring this part of the manuscript.

3.Results 3.6-3.7: The manuscript suggests that the MIF signaling pathway is crucial for the development of the TME in HBV-related HCC and that DNTTIP2 may modulate this pathway to induce immunosuppression. High expression of DNTTIP2 is validated in HepG2 cells and HCC tissue samples. However, the role of the MIF signaling pathway itself has not been experimentally validated. It is recommended to test key molecules in the MIF signaling pathway, such as MIF and CD74, in HepG2 cells and human HCC samples to further substantiate these findings.

4.Functional Validation: While the manuscript validates high expression of DNTTIP2 in HBx-expressing HepG2 cells through qPCR and Western blotting, it lacks direct in vivo functional validation. It is recommended to conduct experiments in an HCC mouse model by knocking down or overexpressing DNTTIP2, observing its effects on tumor growth, immune cell infiltration, and response to immunotherapy, to further validate its functional role in the TME.

5.Introduction: In the third paragraph of the introduction, a more comprehensive review of the related literature would strengthen the discussion. For instance, the following studies should be referenced and cited: PMID: 36939783, PMID: 39368944, etc. A more thorough overview of current research and treatment for HBV should be provided.

6.Minor Grammatical Issues: There are some minor grammatical errors in the manuscript. For example, the sentence: "Immune analysis revealed that high-risk patients had lower survival rates, reduced anti-tumor immune cell infiltration, poorer immunotherapy responses, and increased immune evasion." could be rephrased for clarity. Consider revising it to: "Immune analysis revealed that high-risk patients exhibited lower survival rates, decreased infiltration of anti-tumor immune cells, poorer responses to immunotherapy, and increased immune evasion."

Reviewer #4: In this report J Zhong and coworkers exposed the outcome of analyses conducted in silico and in vitro to identify Hepatitis B virus (HBV) X protein (HBx)-responsive genes influencing the prognosis of patients suffering from a HBV-associated hepatocellular carcinoma (HCC). To this aim the authors studied bulk and single cell gene expression from public databases. They also analyzed the expression of a transiently HBx-expressing cell line (HepG2). Finally, they explored by immunohistochemistry the expression of target genes on paraffin-embedded tissue blocks.

The article has some merits but suffers from crippling shortcomings. The most important one is undoubtedly the problem of figure resolution. Most of them if not all of them are not exploitable for an honest and benevolent observer.

The second problem concern some inferences stemming from in silico analyses that are not always clearly explained or transparent.

Other issues:

Abstract : “Immune analysis revealed that high-risk patients had lower survival rates” this is a pleonastic statement.

Was MIF expression also correlated with that of DNTTIP2 in bulk RNA seq data?

M&ms: Were all 374 HCC analyzed obtained from HBV-infected patients?

Single-cell transcriptome: Mention that it comes from publicly available data and provide the accession number.

Workflow: HepG2 does not appears on the workflow. It is difficult to understand its importance in the whole analytic process.

Furthermore it is usure that these experiments on HepG2 are validating exvivo data or if they are simply relevant. We do not know the expression status of the HBx hub genes HepG2.

Why doing LASSO Cox regression of four genes only and not on all hub genes?

Immune cell infiltration: Figures 5A, C, D are not relevant to this paper.

Figure 5B: it is difficult to see the negative correlation. It seems that there is a lack of any correlation. Please provide the name of cell categories suspected to promote tumor progression and that positively correlated with the 4 hub gene expression. Are CD4 lymphocytes included ?

RPF2 display an inverse correlation with VEGFR: this is somewhat counterintuitive for a gene decreasing survival time for HCC patients. It is absolutely not commented. Why?

Figure 6A is not obvious as there is a large overlap between high risk and low risk groups.

Figure 6B seems to be a correlation matrix between cells population. We do not see where is the risk score exactly.

Figure 6HK: all IPS scores are lower in high-risk tumors even fucntionnally opposite scores such as CTLA4+/PD1+ and CTLA4-/PD1-. What should we conclude? It is very confusing. In general this series of figure are very difficult to understand with the minimalist legend provided.

How many patients have been studied by ScRNA-seq?

The cell chat figure is not explained. It looks as a PCA analysis but we hardly see any “communication” between cell populations

Honestly, the impact of HBx on protein staining in H&E is quite difficult to define (increase? Decrease?)

Reviewer #5: The authors have constructed a prognostic model based on seven HBx-related genes, utilizing four key genes to develop a prognostic risk score signature. Immune analysis indicates that high-risk patients exhibit lower survival rates and other associated issues. Additionally, DNTTIP2 may play a pivotal role in shaping an immunosuppressive tumor microenvironment (TME), which represents a novel concept. However, there are several issues in the manuscript that warrant further consideration:

The manuscript uses GSE276530 as the primary dataset for gene selection, yet it lacks background information regarding this dataset. Moreover, the specific cell treatment processes and experimental methods are not detailed, which raises concerns about the reliability of the data.

There are several grammatical issues throughout the manuscript, including some abbreviations that are repeated upon first use, which may create confusion for readers.

Although the study mentions the selection of seven hub genes, it states that "the expression levels of these seven genes were significantly upregulated in HCC cells expressing HBx and correlated with poor prognosis in HCC patients." Is there a potential bias in this observation that needs to be addressed?

The authors' description of the methodology for selecting model genes is somewhat vague, and the graphical results are not clearly presented, which may hinder the interpretation of the findings.

The study reference transcription factors and miRNAs, presenting relevant results; however, the authors do not discuss the significance of this portion of the results in the analysis.

**Do you want your identity to be public for this peer review?** For information about this choice, including consent withdrawal, please see our Privacy Policy

Reviewer #1: No

Reviewer #2: No

Reviewer #3: **Yes: ** Chen Ling

Reviewer #4: **Yes: ** Pascal Pineau

Reviewer #5: No

---

## [Author Response · Author response to Decision Letter 1]

15 Apr 2025

To the editors,

First of all, we would like to thank the editors for providing us the opportunity to submit a revised version of our manuscript. We would also like to thank the reviewers for their professional remarks and suggestions. We have read through the comments carefully and have made corrections accordingly. All of the questions from the reviewers were answered one by one. The revisions are marked in red, as shown below. We hope this revision can meet the rigorous and professional requirements of PLOS ONE.

In addition to addressing the reviewers’ comments, we have made the following revisions in accordance with the journal’s requirements:

1.We have formatted the manuscript according to PLOS ONE’s guidelines.

2.We have successfully bound our ORCID iD to the corresponding author’s profile in Editorial Manager.

3.We have provided the original, uncropped, and unadjusted images underlying all blot or gel results as per PLOS ONE’s policy.

4.Captions for the Supporting Information files have been included at the end of the manuscript.

We appreciate the constructive feedback provided and believe that these revisions have enhanced the quality of our manuscript.

Reviewer 1

On behalf of all the contributing authors, I would like to express our sincere appreciations of your constructive comments. These comments are all valuable and helpful for improving our article. Accordingly, we revised the manuscript. All comments were responded one by one and all the revised points were marked by red. Hope the revision can be satisfactory for the strict and professional publication requirement of PLOS ONE.

Q1: In the discussion section, the authors should present a more robust account of the canonical function of DNTTIP2 and possibly what is already known about its role in the TME and tumorigenesis. Since experiments to elucidate the molecular mechanism(s) by which DNTTIP2 may drive HCC is lacking.

Response: First of all, we want to express our appreciation for your wise advice and comments. We have expanded the discussion section to provide a more detailed overview of the canonical function of DNTTIP2. We also included what is currently known about its potential involvement in TME and tumorigenesis. According to your comments, the relevant content in the discussion section has been added as follows:

“DNTTIP2, also known as terminal deoxynucleotidyltransferase interacting factor 2 (TdIF2), plays key roles in both cellular processes and tumorigenesis. It is an acidic protein that interacts with TdT and has been identified as a histone chaperone within the nucleus. It binds to DNA and core histones, and its C-terminal region is rich in acidic amino acids, which are important for its function in chromatin dynamics[50]. In the context of cancer, DNTTIP2 has been shown to have a significant role in regulating the cell cycle, particularly in pancreatic cancer (PDAC). Depletion of DNTTIP2 leads to G1 arrest in MIA-PaCa-2 cells and G2 arrest in PK-1 cells, with a corresponding downregulation of cell cycle regulators such as SATB1, CDK6, and CDK1. However, DNTTIP2 depletion does not induce apoptosis, highlighting its more direct role in cell cycle regulation rather than in promoting cell death[51]. Additionally, DNTTIP2 has been shown to influence ribosomal RNA (rRNA) gene transcription by interacting with human ribosomal RNA promoters and promoting hrDNAP activity. Through its interaction with the histone acetyltransferase Tip60, DNTTIP2 inhibits Tip60’s acetylation activity and co-localizes with Tip60 in the nucleolus. This interaction also promotes acetylation of the upstream binding factor (UBF), a key regulator of rRNA synthesis[50]. DNTTIP2 plays a crucial role in both TME and tumorigenesis. It promotes M2 tumor-associated macrophages (TAMs) activation and angiogenesis, processes that contribute to tumor progression. Additionally, DNTTIP2 is involved in the IL6/JAK/STAT3 signaling pathway, which is key to tumor growth and immune evasion. The amplification of DNTTIP2 is associated with poor prognosis in glioma patients, and its hypermethylation is also linked to unfavorable outcomes, highlighting its potential as a biomarker and therapeutic target[52].” (Page 31-32, Lines 627-655)

Q2: The authors stated that “Between January 2020 and August 2024, HCC tissues from 3 HBV-infected HCC patients and their corresponding adjacent non-tumor tissues, as well as HCC tissues from 3 HCC patients without HBV infection and their corresponding adjacent nontumor tissues, were collected at the Eighth Affiliated Hospital of Sun Yat-sen University” but representative images of IHC staining of DNTTIP2 in the adjacent non-tumor tumor tissues are not shown. Could the authors provide that?

Response: Thank you for your valuable feedback. We apologize for the omission of representative images of DNTTIP2 IHC staining in the adjacent non-tumor tissues. We have now included these images in the supporting information of the revised manuscript, as the primary focus of the manuscript is on HCC tissues. Our results show that DNTTIP2 is indeed highly expressed in the adjacent non-tumor tissues of HBV-infected patients, similar to its expression in HCC tissues. We have also provided the relevant images below for your reference:

S2 Fig. Representative images of H&E staining and IHC staining of DNTTIP2 protein in HBV-negative and HBV-positive adjacent non-tumor tissues.

Reviewer 2

On behalf of all the contributing authors, I would like to sincerely thank you for your constructive comments. Your feedback has been invaluable in enhancing the quality of our manuscript. In response, we have revised the manuscript accordingly. Each comment has been addressed individually, and all revisions are highlighted in red. We hope that these changes meet the rigorous and professional standards required by PLOS ONE.

Q1: Page 17, line 208: Although 2^-ddCt is a common technique, it assumes that the amount of sample is doubled at each cycle, and this is not always the case. It would be convenient to use a standard calibration curve using serial dilutions of a pull of samples. It is also important to check more than one normalizer and use GeNorm or Normfinder to find a normalization factor. This reviewer recommends using 2 or 3 in total or at least demonstrating that GAPDH is a good normalizer by showing the Ct's (which should be fairly similar across samples).

Response: Thank you for your valuable suggestions regarding the normalization methods. We highly appreciate your recommendation to use a standard curve and multiple normalization factors, and we will consider this as a direction for improvement in future studies. Regarding our current qPCR analysis method, our team has been using the 2^-ddCt method. While we plan to explore more advanced normalization techniques in the future, our current method assumes that the amplification efficiency of the target gene and the reference gene is similar. Both the target gene and GAPDH, as the reference gene, exhibit similar slopes during the exponential phase of the amplification curves, indicating consistent amplification efficiency. Although we acknowledge the need for more advanced methods, we are currently able to apply the 2^-ddCt method with reasonable confidence. Regarding GAPDH as a reference gene, we have validated its stability through multiple experiments. GAPDH has been a long-standing and reliable internal control in our lab, and it has been widely used and validated in several relevant studies, particularly in liver cancer and other tumor-related research. We also performed a comparative analysis of the Ct values of GAPDH across different samples, and the results showed consistent expression across all groups, confirming its stability. Additionally, GAPDH’s stability and consistency have been consistently validated under various experimental conditions, further supporting its suitability as a reference gene in our current study. We understand the importance of selecting appropriate normalization factors for ensuring data reliability, and we will continue to refine and validate these methods in future research. We appreciate your insightful feedback, which provides valuable guidance for the direction of our future studies. The table below shows the Ct values of GAPDH across different samples.

the Ct values of GAPDH the Ct values of GAPDH

NC-1 15.01 HBx-1 15.11

NC-2 15.04 HBx-2 15.14

NC-3 14.22 HBx-3 15.12

NC-4 14.17 HBx-4 14.17

NC-5 14.13 HBx-5 14.17

NC-6 14.15 HBx-6 14.16

NC-7 14.38 HBx-7 14.39

NC-8 14.44 HBx-8 14.41

NC-9 14.42 HBx-9 14.37

Q2: Page 19, line 267: Here the threshold is set to |logFC|>0.5 (same for figure 1). In the Methods (page 14, line 118) the values were |log2FC|>1, which makes more sense as it assumes a FC greater than 2-fold. Is the value set here log10? Anyway, there is a mistake that needs to be corrected somewhere.

Response: Thank you for your careful review. You are absolutely right — the inconsistency in the fold change threshold was due to a writing error. The correct threshold used throughout our analysis was |log2FC| > 0.5, and we have now revised the manuscript to ensure consistency in both the Results and Methods sections. We appreciate you pointing this out. The corresponding content of the manuscript has been modified as follows:

“All identified DEGs met the criteria of P < 0.05 and |log2 (Fold-change)| > 0.5. ” (Page 9, Lines 189-190)

“Based on three algorithms, a differential analysis was conducted on HepG2 cells expressing HBx compared to control group (|log2FC| > 0.5, P < 0.05). ” (Page 20, Lines 394-396)

Q3: Page 20, line 283: FOR ALL THE FIGURES’ FOOTNOTES. Knowing that this is not the final format of the paper, this reviewer suggests for the next submission to put all footnotes together, at the end of the text or with the figures or as supplementary material. It is very difficult to find them when you are reading the text and looking at the figures until you realize that they are at the end of each part.

Response: Thank you for your valuable suggestion regarding the footnotes. We understand that having footnotes scattered throughout the text can make it difficult to follow. In response to your comment, we have now gathered all the footnotes together and placed them at the end of the manuscript for better clarity and ease of reference. We appreciate your feedback and believe this change will improve the readability of the paper.

Q4: Page 20, line 287: More information is needed for correct interpretation of Figures 2E-G. The outer boxes, with the dots, are divided into four regions or levels where the dots are drawn. What do those levels mean? The inner boxes, where the z-score color is used, have different sizes. Why?

Response: Thank you for your thoughtful comment regarding the interpretation of Figures 2E-G. We are happy to provide further clarification:

In these figures, the outer boxes represent the distribution of differentially expressed genes in each pathway, based on their log fold change (log2FC) values. The red dots indicate upregulated genes, while blue dots indicate downregulated genes. The further a dot is from the center, the greater the absolute log2FC value it has. The four regions in each outer box correspond to different log2FC intervals, and these intervals are scaled according to the log2FC distribution within each individual pathway.

The inner boxes represent the overall activity trend of the pathway based on the z-score. The color intensity reflects the magnitude of the z-score. Additionally, the size of the inner box corresponds to the statistical significance (p-value) of the pathway enrichment - larger boxes indicate more significant pathways.

We have revised the figure legend in the manuscript to explain these visual elements more clearly and help readers interpret the figures correctly:

“E-G The enriched GO terms of DEGs: (E) Biological Process, (F) Cellular Component, and (G) Molecular Function. Red and blue dots represent upregulated and downregulated genes based on log2FC. Inner box color indicates pathway z-score, and box size reflects enrichment significance.”

Q5: Page 20, line 293: Although the text is fine and I would keep it, Figure 3A is too complicated and, in this reviewer's opinion, does not provide much information (Figure 3B is sufficient to understand the results). I suggest moving Figure 3A to supplementary figures.

Response: Thank you for your constructive feedback. We understand your concern regarding the complexity of Figure 3A. After considering your suggestion, we have moved Figure 3A to the supplementary figures section, as Figure 3B alone provides sufficient information for understanding the results. We believe this will enhance the clarity of the main text. Thank you for your helpful input. The revised Figure 3 and the newly added Figure S1 are shown below:

Fig 3. PPI Network Analysis, Hub DEGs Identification, and Survival Analysis in HCC.

A A key cluster with 12 genes by MCODE based on the PPI network. B KM-plot of prognostic HBx-related hub genes based on TCGA cohort. C KM-plot of prognostic HBx-related hub genes based on GEPIA. D The expression of prognostic hub DEGs in the normal and tumor tissues (***P < 0.001, (****P < 0.001). E The correlation between the seven prognostic hub DEGs using Spearman analyses. Positive correlation was marked with red. Colors represent different ranges of Pearson correlation coefficients (e.g., dark red indicates R = 0.8 - 1.0). The colored areas are categorized based on correlation strength, not proportionally scaled to the R value.

S1 Fig. Characterization of HBx-Related DEGs and MIF Signaling Interactions in HBV-Related HCC.

A PPI network of HBx-related DEGs. B The number of interactions and interaction strength between different celltypes in HBV-related HCC tissues. C The interaction strength of MIF signaling pathway between different celltypes in HBV-related HCC tissues. D Signaling from malignant cells to other cells in the MIF signaling pathway. A thicker line indicates a higher interaction strength or a greater number of interactions.

Q6: Page 21, line 305: The upstream transcription factor analysis is a good idea, but these results seem to be no longer used. Does this figure and result add anything to the paper? This reviewer would remove this figure.

Response: Thank you for your valuable comment. After reconsidering the relevance of the upstream transcription factor analysis, we agree that it does not significantly contribute to the main findings of the paper. Therefore, we have removed this figure from the manuscript. The revised Figure 3 is shown above. We appreciate your suggestion, which has helped us streamline the content.

Q7: Page 21, line 312: More information is needed for correct interpretation of the 3F figures. The color used in the graph is according to the Pearson R value, but is the colored area also proportional to this value?

Response: Thank you for your comment. The different colored areas in Figure 3F correspond to manually defined ranges of Pearson correlation coefficients (e.g., dark red represents R values between 0.8 and 1). These color assignments are categorical and intended to visually differentiate levels of correlation, but they are not proportionally scaled based on mathematical relationships. We have clarified this in the figure legend accordingly:

“Colors represent different ranges of Pearson correlation coefficients (e.g., dark red indicates R = 0.8 - 1.0). The colored areas are categorized based on correlation strength, not proportionally scaled to the R value.”

Q8: Page 21, line 314: Since there is no negative correlation in Figure 3F, the blue color comment is not necessary. I would leave just the red color comment.

Response: Thank you for your suggestion. We agree with your comment and have removed the blue color comment. Only the red color comment, which indicates a high level of positive correlation, has been retained in the revised figure legend:

“Positive correlation was marked with red.”

Q9: Page 22, line 336: More info is needed for the correct interpretation of the Figure 4C. What do the columns of the histogram represent?

Response: Thank you for your insightful comment. We have revised the figure legend to clarif

---

## [Decision Letter · Decision Letter 1]

Hepatitis B virus X protein (HBx)-mediated immune modulation and prognostic model development in hepatocellular carcinoma

PONE-D-24-59329R1

Dear Dr. Zhai,

We’re pleased to inform you that your manuscript has been judged scientifically suitable for publication and will be formally accepted for publication once it meets all outstanding technical requirements.

Kind regards,

Li Shen

Academic Editor

PLOS ONE

Additional Editor Comments (optional):

Reviewers' comments:

Reviewer's Responses to Questions

**Comments to the Author**

Reviewer #1: All comments have been addressed

Reviewer #2: All comments have been addressed

Reviewer #3: All comments have been addressed

2. Is the manuscript technically sound, and do the data support the conclusions?

Reviewer #1: (No Response)

Reviewer #2: Yes

Reviewer #3: Yes

3. Has the statistical analysis been performed appropriately and rigorously?

Reviewer #1: (No Response)

Reviewer #2: Yes

Reviewer #3: Yes

4. Have the authors made all data underlying the findings in their manuscript fully available?

Reviewer #1: (No Response)

Reviewer #2: (No Response)

Reviewer #3: Yes

5. Is the manuscript presented in an intelligible fashion and written in standard English?

Reviewer #1: (No Response)

Reviewer #2: Yes

Reviewer #3: Yes

Reviewer #1: (No Response)

Reviewer #2: (No Response)

Reviewer #3: (No Response)

**Do you want your identity to be public for this peer review?** For information about this choice, including consent withdrawal, please see our Privacy Policy

Reviewer #1: No

Reviewer #2: No

Reviewer #3: **Yes: ** Chen Ling

---

## [Editor Report · Acceptance letter]

PONE-D-24-59329R1

PLOS ONE

Dear Dr. Zhai,

I'm pleased to inform you that your manuscript has been deemed suitable for publication in PLOS ONE. Congratulations! Your manuscript is now being handed over to our production team.

Kind regards,

on behalf of

Dr. Li Shen

Academic Editor

PLOS ONE